# Do Micronutrient and Omega-3 Fatty Acid Supplements Affect Human Maternal Immunity during Pregnancy? A Scoping Review

**DOI:** 10.3390/nu14020367

**Published:** 2022-01-15

**Authors:** Gail Rees, Louise Brough, Gustavo Moya Orsatti, Anna Lodge, Steven Walker

**Affiliations:** 1School of Biomedical Sciences, University of Plymouth, Drake Circus, Plymouth PL4 8AA, UK; 2School of Food and Advanced Technology, Massey University, Palmerston North 4442, New Zealand; 3P&G International Operations SA, 1213 Geneva, Switzerland; moyaorsatti.g@pg.com; 4St Gilesmedical, London WC2H 8LG, UK; anna.lodge@stgmed.com (A.L.); steven.walker@stgmed.com (S.W.)

**Keywords:** pregnancy, immunity, supplementation, micronutrients, scoping review

## Abstract

Maternal dietary micronutrients and omega-3 fatty acids support development of the fetal and neonatal immune system. Whether supplementation is similarly beneficial for the mother during gestation has received limited attention. A scoping review of human trials was conducted looking for evidence of biochemical, genomic, and clinical effects of supplementation on the maternal immune system. The authors explored the literature on PubMed, Cochrane Library, and Web of Science databases from 2010 to the present day using PRISMA-ScR methodology. Full-length human trials in English were searched for using general terms and vitamin A, B12, C, D, and E; choline; iodine; iron; selenium; zinc; and docosahexaenoic/eicosapentaenoic acid. Of 1391 unique articles, 36 were eligible for inclusion. Diverse biochemical and epigenomic effects of supplementation were identified that may influence innate and adaptive immunity. Possible clinical benefits were encountered in malaria, HIV infections, anemia, Type 1 diabetes mellitus, and preventing preterm delivery. Only limited publications were identified that directly explored maternal immunity in pregnancy and the effects of micronutrients. None provided a holistic perspective. It is concluded that supplementation may influence biochemical aspects of the maternal immune response and some clinical outcomes, but the evidence from this review is not sufficient to justify changes to current guidelines.

## 1. Introduction

Pregnancy is a demanding process. The mother needs to maintain her own homeostasis while simultaneously supporting placental function and nurturing a healthy fetus. From an immunological perspective, she is faced with two significant challenges: defending her body against pathogens and tolerating fetal antigens. 

Healthy women usually carry a semi-allogeneic fetus without immune rejection, despite expression of different paternal antigens by the fetus [1]. Traditionally, the immunological effects of pregnancy were considered to be suppressive, with pregnant women at greater risk of infection [2]. The process was viewed as a continuum, with T helper (Th) 2 immunity predominating to protect the fetus at the expense of Th1 proinflammatory cytokines [3,4].

This is now considered an oversimplification. The maternal-fetal immunological relationship seems to be highly complex and bidirectional. Evidence suggests that pregnancy involves three distinct immunological stages [5]. During the first trimester, there is a strong inflammatory response consistent with the human placenta being hemochorial, meaning that fetal tissue is penetrating the endometrium to interact with maternal blood. Thus, implantation and placentation resemble an open wound. 

On reaching the second trimester, the immunological profile switches to an anti-inflammatory state as the placenta and fetus continue to develop. Finally, with the mother approaching delivery in the third trimester, a new proinflammatory environment is needed to promote induction [5]. These changes are linked to alterations in the uterine lymphocyte population, notably natural killer cells, which have an important role in placental vascular remodeling and trophoblast invasion [2,6,7]. Thus, pregnancy represents a unique immunological state that requires careful control. 

Animal studies suggest that micronutrient supplementation plays an important role in optimizing immunity during pregnancy and peripartum for both the mother and offspring [8,9,10,11,12]. Similarly, the medical and scientific community are learning more about the role of micronutrients in human pregnancy [13]. 

In a WHO report published in 2000, deficiencies of vitamin A, zinc, iodine, and iron were identified as some of the most serious risk factors for global health. Highlighted consequences were increased susceptibility to infections, metabolic disorders, and impaired physical development [14]. Some micronutrients are thought to be particularly important for immunocompetence. Vitamin A, for example, plays a major role in maintaining epithelial and mucus integrity and developing the immune system. Here, it plays a regulatory role in the cellular and humoral response, notably the formation of T cells [13,15]. In the fetus, vitamin A deficiency impairs B cell differentiation and the development of lymphoid tissue [16]. 

Selenium is a key factor for leukocyte function: it protects leukocytes from oxidative stress, is involved in T cell proliferation, and participates in humoral immunity [17,18]. In pregnant women, selenium deficiencies may induce gestational complications and damage the fetal nervous and immune systems [19]. Much has been written about vitamin D, which is now recognized to have a range of effects. Most leukocytes carry the vitamin D receptor (VDR) on their surface, suggesting a widespread role in immune regulation. Though technically not essential, omega-3 fatty acids alphalinolenic acid (ALA), eicosapentaenoic acid (EPA), and docosahexaenoic acid (DHA) have numerous important functions in the body and a potential role in modulating immune response during pregnancy [20].

Most trials on peripregnancy supplementation have concentrated on potential benefits for the fetus and newborn, rather than taking a holistic approach. Aspects considered to date include improving placental function and colostrum quality, reducing atopy, and enhancing neural development. However, little is known about whether the mother herself derives any benefit. The purpose of this scoping review is to look for evidence of the effects of key micronutrient and omega-3 supplementation on maternal immunity and whether these agents result in clinical benefit. Also of interest is mapping key concepts, main sources of information, and study types, and clarifying the definitions and boundaries of the topic [21].

## 2. Materials and Methods

The study protocol was registered with the Open Science Framework (OSF CM3XA) [22]. 

A scoping review was performed using the PRISMA extension for scoping reviews (PRISMA-ScR) guidelines advocated by Tricco et al. [23]. 

The PICO question was “In pregnant women (P), what is the effect of dietary supplementation with one or more micronutrients and/or omega-3 fatty acids (I), compared with baseline or control subjects not receiving supplementation (C) on parameters of maternal immunity (O)?”.

### 2.1. Information Sources

The authors searched PubMed, the Cochrane Library, and Web of Science using the strategy in Table 1.

These search terms were selected based on the experience of 2 expert academic nutritionists (GR and LB). The term ‘probiotics’ was also included with the aim of identifying papers that describe the effects of multicomponent supplementation. The search period was chosen following a preliminary exploration of the topic and in anticipation that this would result in a manageable dataset that is likely to encompass the most relevant publications.

### 2.2. Eligibility Criteria

Inclusion criteria: •Articles in English published in established peer-review journals between 1 January 2010 and 26 April 2021;•Full-length articles describing potential biochemical, epigenomic, and clinical effects of micronutrient and omega-3 long-chain fatty acid supplementation on immunity in pregnant women;•Longitudinal trials/randomized, controlled trials involving dietary supplementation in pregnant women at any stage of gestation, where comparison is made with their status at enrollment and/or with a control population;•Participants must have been assessed clinically and/or biochemically at enrollment and at one or more timepoints thereafter;•Study participants with gestational comorbidities were not excluded from this review.

Exclusion criteria: •Articles focusing on fortification or the effects a particular food rather than supplementation; Foods promoted for well-being typically contain a range of nutritional components, making the identification of benefits associated with individual micronutrients problematic. Similarly, it is difficult to separate the effects of fortification from the elements of the product being added to. •Animal studies;•Articles that concentrate on potential benefits for the fetus and newborn (e.g., effects on placental function, colostrum quality, atopy, and neural development);•Reviews, trial protocols, letters, conference reports, withdrawn manuscripts, and non-English publications.

### 2.3. Risk of Bias

Risk of bias was not assessed. Unlike for systematic reviews, the process of assessing the risk of bias and synthesizing findings from individual studies to generate summary findings is not mandatory [24].

### 2.4. Synthesis of Results

Titles and abstracts were collected in EndNote (Clarivate Analytics, Philadelphia, PA, USA) and screened independently by 2 reviewers (AL and SW) with the support of 2 further reviewers (KL and LD). Duplicates were removed. Full-text versions of all selected publications were then examined by 2 reviewers (AL and SW). Differences of opinion were resolved by 2 expert academic nutritionists (GR and LB). Information was extracted from selected papers and tabulated. 

## 3. Results

A search of PubMed resulted in the identification of 300 potential publications, the Cochrane Library in 473 publications, and Web of Science in 1138 publications (Figure 1). After removal of duplicates, there were 1391 publications to be screened. Examination of the titles and abstracts resulted in 71 full-length articles being selected for examination, of which 36 form the substance of this scoping review. Involved research centers were predominantly in the USA (*n* = 11), Europe (*n* = 10), and Africa (*n* = 8). Study periods ranged from nine weeks’ gestation to six months post-partum.

Selected publications frequently explore the effects of micronutrient supplementation alone or in combination. Others were found to focus on human pathophysiology (e.g., immunity after vaccination or a disease process such as malaria or HIV). Consequently, a particular study may be mentioned more than once under different headings.

### 3.1. Vitamin A

Vitamin A is considered important for normal maternal immune function as well as to improve host defense against malaria and reduce the risk of adverse pregnancy outcomes (see below).

In a randomized, controlled, clinical trial of oral vitamin A supplementation, 112 pregnant Bangladeshi women in their second trimester (11–14 weeks) with predominantly low plasma retinol concentrations were assigned to receive either oral 10,000 IU vitamin A or a placebo weekly, until six months postpartum [25]. During the third trimester, participants were inoculated with a single dose of inactivated pandemic H1N1-(swine flu) influenza vaccine. Hemagglutination-inhibition (HAI) titers were measured in cord, infant, and maternal blood samples. Ahmad et al. found that supplementation during pregnancy and postpartum increased breastmilk vitamin A concentrations and enhanced prenatal H1N1-vaccine responses. No harmful effects were reported. The researchers postulate that effects on transplacental antibody transfer may depend on the time of vaccination.

Darling et al. investigated the effects of daily oral supplementation with either vitamin A or zinc, alone or in combination, on placental malaria and pregnancy outcomes. The group conducted a randomized, double-blinded, placebo-controlled trial in 2500 HIV-negative primigravid or secundigravid pregnant women in Dar es Salaam, Tanzania [26]. Participants were equally randomized (625 × 3) to receive 2500 IU of vitamin A, 25 mg of zinc, a combination of these two micronutrients, or a placebo daily until delivery. Supplement administration commenced in the first trimester. 

Placental samples were obtained at delivery in 1404 participants and data on birth outcomes in 2434 of 2500 participants. Secondary outcomes included small for gestational age (SGA) births and prematurity. Those who received zinc were found to be less likely to exhibit histopathology-positive placental malaria (relative risk [RR] 0.64; 95% confidence interval [CI] 0.44–0.91). Neither nutrient influenced polymerase chain reaction-positive malaria, SGA, or prematurity. No safety concerns were reported.

Below, the work of Sun et al. (2010) [27] on iron plus retinol supplementation is referred to. Also, a randomized, placebo-controlled trial by Olofin et al. that examined whether daily multivitamin supplementation (vitamin B complex, C, and E) or vitamin A altered malaria incidence in HIV-infected Tanzanian women of reproductive age is included [28].

### 3.2. Vitamin B12

Vitamin B12 deficiency is common in pregnancy, and is associated with lower birth weight (birth weight <2500 g) and preterm births (length of gestation <37 weeks). There is limited information regarding the effects of this micronutrient on immune function. In a blinded, placebo-controlled trial, Siddiqua et al. evaluated the effects of pre- and postnatal supplementation on biomarkers of B12 status and vaccine-specific responses to the H1N1 influenza vaccine in Bangladeshi mothers and infant effects [29]. 

Pregnant women (*n* = 68, age 18–35 years, 11–14 weeks of gestation, hemoglobin <110 g/L,) were randomized to receive B12 (250 μg/day) or a placebo during pregnancy and for three months’ postpartum, along with 60 mg iron plus 400 μg folate. Women were immunized with H1N1 vaccine at 26 to 28 weeks’ gestation. Blood samples from mothers (baseline, 72-h post-delivery, three months postpartum), neonates, and infants (three months of age) were analyzed for hemoglobin, B12, methylmalonic acid (MMA), total homocysteine (tHcy), serum ferritin and serum transferrin receptor, C-reactive protein (CRP), and alpha-1-acid glycoprotein (AGP). Vitamin B12 concentrations were measured in breast milk, while H1N1-specific antibodies were analyzed in plasma, colostrum, and breast milk. The researchers found that supplementation increased B12 in plasma, colostrum, and breast milk (*p* < 0.05), and lowered MMA in mothers, neonates, and infants at three months of age (*p* < 0.05). Compared with placebo, administration of B12 significantly increased H1N1-specific IgA responses in plasma and colostrum in mothers while reducing the proportion of infants with elevated AGP and CRP. The latter suggests that supplementation may reduce inflammatory responses in infants. No supplementation-related harmful effects were reported.

### 3.3. Vitamin C and E

A study from Belfast explored the potential benefits of vitamin C (1000 mg) and E (400 IU) supplementation administered daily between 8 and 22 weeks of gestation until delivery in type 1 diabetes, where rates of preeclampsia are known to be two to four times higher than in non-diabetic pregnancy [30]. It is suggested that the underlying mechanism is due to increased oxidative stress in both type 1 diabetes and preeclampsia.

The researchers examined placental samples from 57 mothers with various clinical profiles enrolled in a previous antioxidant supplementation study in diabetes (DAPIT). A range of parameters were examined, including placental protein tyrosine nitration and p38-mitogen-activated protein kinase α (p38-MAPKα), extracellular-signal regulated kinase (ERK), and c-Jun NH2-terminal kinase (JNK), using immunohistochemistry and an enzyme-linked immunosorbent assay (ELISA). Results demonstrated immunopositivity in normotensive diabetic placenta, suggesting that tyrosine nitration had occurred. Nitrotyrosine immunostaining was found in placebo-treated normotensive, gestational hypertensive, and preeclampsia placenta, with no significant differences between groups. Further, p38-MAPKα, ERK, and JNK were not directly involved in the pathogenesis of type 1 diabetic preeclampsia and were unaffected by vitamin supplementation [30]. 

Research regarding the potential role of vitamin D in reducing the risk of preeclampsia is presented below.

### 3.4. Vitamin D

The primary role for vitamin D was originally thought to be in supporting bone and calcium metabolism. It is now realized that this micronutrient is essential for growth and differentiation, as well as supporting both innate and adaptive immunity [7]. The antiinflammation effects of vitamin D are considered important, as inflammation has been linked to poor birth outcome. It is noted in a study of healthy, pregnant African-American women that serum 25-hydroxyvitamin D (25-OHD) concentrationsdemonstrated seasonal variation and were inversely associated with second-trimester biomarkers of inflammation [31].

Vitamin D receptors have been widely identified [7]. During peripregnancy, calcitriol synthesized in the ovary, decidua, endometrium, and placenta influences the differentiation and proliferation of immune system modulators (e.g., B cells, T cells, Th [helper] cells, Treg [regulatory] cells, macrophages, and dendritic cells), interleukin expression, and antimicrobial responses. These processes affect all stages of pregnancy, including fertilization, implantation, and maintenance of gestation. Inadequate vitamin D levels are associated with recurrent implantation failure (RIF), recurrent pregnancy loss, the outcome of assisted reproduction technology, and complications such as preeclampsia. These effects may be due to cytokine patterns and an imbalance of Th1 and Th2 activity. 

The current literature search identified eight trials that sought to examine the potential role of vitamin D supplementation. Six of these broadly explore biochemical parameters, while two have a clinical focus. These are summarized in Table 2.

Low vitamin D levels have been proposed as a risk factor for preeclampsia. In a reanalysis of data from a large randomized study (Vitamin D Antenatal Asthma Reduction Trial; VDAART), Mirzakhani et al. examined the effects of early administration (from weeks 10–18) of recommended (400 IU/day) vs. high-dose (4400 IU/day) vitamin D supplementation, as well as associated gene expression profiles in the peripheral blood of 47 patients developing preeclampsia [32] (Table 2).

Overall, there were no differences between treatment (*n* = 408) or controls (*n* = 408) in the incidence of preeclampsia (8.08 vs. 8.33%, respectively; RR 0.97; 95% CI 0.61–1.53). However, blood 25-hydroxyvitamin D (25-OHD) concentrations ≥30 ng/mL at trial entry and in late pregnancy were associated with a lower risk of preeclampsia. 

Interestingly, there was significantly greater differential expression of 348 vitamin D–associated genes (158 upregulated) in women who developed preeclampsia. This suggests the emergence in early pregnancy of a distinctive immune response. Further analysis showed the presence of several highly functional modules mapped to systematic inflammatory and immune responses, including several nodes with a high connectivity. 

Patterns of gene expression alter during pregnancy, and these changes are poorly understood. A further analysis of the VDAART trial examined longitudinal changes in the maternal transcriptome. RNA from the peripheral blood of 30 pregnant women randomized to receive either low- or high-dose vitamin D supplementation (400 vs. 4400 IU, daily) was studied at enrollment (10–18 weeks’ gestation) and again during the third trimester (32–38 weeks) [33].

Data from this study demonstrated 5839 differentially expressed genes (false discovery rate < 0.05), both up and downregulated, clustered into 14 co-expression modules. Among these modules, two significantly correlated with maternal vitamin D levels. Further analysis of these modules demonstrated enhancement of genes involved in immune defense, extracellular matrix reorganization, and notch signaling and transcription. The researchers concluded that maternal vitamin D levels influence transcriptional profiles and suggest that these gene changes may in turn affect fetal immunity and reduce allergic sensitization.

In another epigenomic study, Anderson et al. studied how vitamin D supplementation may affect DNA methylation by means of a randomized, controlled pilot study in pregnant and lactating women and their breastfed offspring [34].

Pregnant women received either a standard recommended daily dose of vitamin D3 (400 IU; control group *n* = 6) or a higher dose (3800 IU; intervention group *n* = 7) daily from the late-second trimester through to four to six weeks postpartum. Epigenome-wide DNA methylation was quantified in leukocytes collected from mothers at birth, and both mothers and their infants at four to six weeks postpartum. 

The researchers confirmed that high-dose maternal vitamin D supplementation (intervention group *n* = 7; 3800 IU/day) from late-second trimester through to four to six weeks postpartum altered DNA methylation in mothers and breastfed infants compared with controls (*n* = 6; 400 IU/day). There were variable gains and losses. At birth, intervention-group mothers showed DNA methylation gains at the 76 cystine-guanine (CpG) dinucleotide and losses at 89 CpG dinucleotides. The associated gene clusters map to cell migration/motility and cellular membrane function. By comparison, at postpartum, maternal methylation gains, which are biologically relevant for cadherin signaling and immune function, were seen at 200 and a loss at 102 CpGs. Differences in DNA methylation were also seen in breastfed infants of intervention mothers at four to six weeks.

Recurrent miscarriage (RM; defined as ≥2 consecutive miscarriages before the 20th week of gestation) affects 2–5% of pregnant women. The etiology is unknown, but is widely considered to have an immunological basis. It is known that those at risk of RM have a higher Th17/T regulatory (Treg) ratio in their peripheral blood. Lymphocyte immune therapy (LIT) is widely used to prevent RM. It can be seen from the above publications that 1a, 25-dihydroxy-vitamin-D3 has immunosuppressive properties. A double-blinded, placebo-controlled study by Rafiee et al. examined the effects of LIT alone or in combination with vitamin D supplementation on Th17/Treg cells in women at risk of RM [35]. The researchers found that the combination therapy decreased both the frequency of Th17 cells and the Th17/Treg ratio compared with the control group (*p* < 0.05) and suggest that vitamin D may be a therapeutic candidate in RM. 

The effects of vitamin D on innate and adaptive immunity and inflammation during pregnancy were also investigated by Zerofsky et al. [36]. The researchers conducted a randomized, double-blinded, controlled trial among 57 pregnant women looking at two doses of vitamin D (400 IU and 2000 IU/day) administered from <20 weeks to delivery. A range of biochemical and clinical outcomes were assessed.

The higher dose of supplement significantly increased maternal vitamin D status compared with baseline. At 36 weeks, a 2000 IU dose resulted in significantly more interleukin-10* regulatory CD4* T cells. There were also potential nonsignificant benefits in terms of a reduction in maternal diastolic BP and higher birth weights.

One reason for a lack of effect in many studies of vitamin D supplementation may be the timing of administration. In a recent reanalysis of a randomized study, Khatiwada et al. looked at the longitudinal impact of administering either 400 or 4400 IU vitamin D3 from enrollment at 10–14 weeks’ gestation [37]. The researchers collected data on health, safety, circulating 25(OH)D, and nine immune mediators during each trimester, demonstrating the importance of baseline values. They found an association between baseline 25(OH)D and baseline TGF-β and the second- and third-trimester IFN-γ and IL-2. While there were racial differences, vitamin D supplementation throughout pregnancy did not impact immune mediators at later trimesters. They suggest that supplementing with vitamin D before conception may influence immune-mediator responses during pregnancy.

Finally, this section includes two papers linking maternal health in pregnancy and vitamin D status, which may have an immunological basis. The first by Samimi et al. demonstrated the beneficial effect of administrating vitamin D plus calcium during pregnancy on glycemic status, high-density lipoprotein cholesterol, plasma total glutathione concentration, and blood pressure [38].

The second explores an inverse link between vitamin D status and bacterial vaginosis, which in turn may affect pregnancy outcomes. This was investigated by means of a randomized study of pregnant women receiving either standard-dose vitamin D (400 IU daily; *n* = 191; control group) or high-dose supplementation (4400 IU; *n* = 196; treatment group) [39]. The researchers found that the microbiome was significantly associated with gestational age and race, and that patterns of colonization were associated with 25(OH)D concentration. It is suggested that vitamin D promotes mucosal integrity and/or alters the vaginal environment.

### 3.5. Choline

Pregnancy increases the maternal and fetal demand for choline. Choline is essential for placental function, fetal growth, and brain development [40]. It is a major component of cell and organelle membranes and is required for nerve myelination, synthesis of the neurotransmitter acetylcholine, bile secretion, alveolar surfactant formation, hepatic lipid export, and one-carbon metabolism. Inadequate maternal choline supply to the developing fetus may result in birth defects and impaired cognitive ability. 

There is a strong association between choline supply and the regulation of gene expression (epigenetics). Pregnancy induces physiological adaptations that may involve, or contribute to, alterations in the genomic landscape secondary to its role as a methyl donor. In a controlled feeding study, Jiang et al. sought to explore the relationship between pregnancy, choline, and maternal genomic markers [41]. Healthy pregnant women in their third trimester (*n* = 26, week 26–29 gestation) and non-pregnant (*n* = 21) women were randomized to receive a choline intake that approximated the US Adequate Intake level (480 mg/day) or nearly a double dose (930 mg/day) for 12 weeks. Blood samples were taken on day 0 and at the end of week 12. Leukocytes were assessed for microarray, DNA damage, and measurement of global DNA/histone methylation. Interestingly, the researchers found that healthy, third-trimester pregnant women experienced transcriptional activation of neutrophils in the peripheral innate immune system and elevated leukocyte oxidative DNA damage compared with non-pregnant women. 

These results suggest a pregnancy-associated upregulation of the innate immune system, which may play a role in protecting the mother and fetus against pathogens. Apart from greater histone 3 lysine 4 di-methylation levels in pregnant women receiving the higher dose (*p* = 0.03), choline did not appear to affect any variable under investigation during the third trimester.

### 3.6. Iron and Lactoferrin

Iron deficiency anemia is common in developing countries and predisposes those affected to infections. Iron plays an important role in protecting humans against pathogens. With respect to innate immunity, iron influences the production of antimicrobial effectors such as nitric oxide and hydroxyl radicals by the myeloid cells. By comparison, its role in adaptive immunity is as an essential growth-factor for the clonal expansion of lymphocyte subsets. Whenever there is immune activation (e.g., due to infection), levels of plasma iron typically fall, accompanied by its compartmentalization within the mononuclear phagocyte system.

Sun et al. investigated the effect of iron combined with retinol supplementation on iron status, IL-2 levels, and lymphocyte proliferation in anemic (≥80 Hb < 110 g/L) pregnant women in rural China by means of a two-month double-blinded trial [27]. The researchers randomized 186 participants into four groups. Group I (*n* = 47) received 60 mg ferrous sulphate daily, IF (*n* = 46) received 60 mg ferrous sulphate plus 0.4 mg folic acid, and IR (*n* = 46) received 60 mg ferrous sulphate plus 2.0 mg retinol and 0.4 mg folic acid. Group C (*n* = 47) was the placebo group. Increases in plasma iron, ferritin, IL-2, and lymphocyte proliferation were seen in the supplementary arms. No harmful effects were reported. It was concluded that iron combined with retinol was more beneficial in improving iron status and lymphocyte proliferation during pregnancy than iron alone.

Many pregnant African women are affected by iron deficiency anemia (IDA). The benefits of antenatal iron supplementation are controversial and there are concerns regarding the risks of increasing the malaria burden through supplementation. Mwangi et al. measured the effects of antenatal iron supplementation by means of a randomized placebo-controlled trial on maternal *Plasmodium falciparum* (*P. falciparum*) infection risk, maternal iron status, and neonatal outcomes in an endemic area of rural Kenya [42].

Participants included 470 women between 15 to 45 years of age with singleton pregnancies, gestational age of 13 to 23 weeks at enrollment, and hemoglobin concentration ≥90 g/L. From enrollment onwards, all women received 5.7 mg iron/day in the form of fortified flour and intermittent preventive malaria treatment as practiced locally. In addition, those randomized to the intervention group received daily iron supplements (60 mg ferrous fumarate, *n* = 237), while the control group received a placebo (*n* = 233). Treatment continued until one month postpartum. The researchers found that the administration of daily iron supplementation resulted in no significant differences in overall maternal *P. falciparum* infection risk compared with the placebo. There was no evidence that iron supplementation caused serious side effects. A positive effect was that iron supplementation resulted in increased birth weight.

Similar findings were reported in a randomized study from Tanzania involving 1500 iron-replete, non-anemic women at or before 27 weeks’ gestation. Etheredge et al. found that antenatal iron supplementation was not associated with an increased risk of placental malaria or other adverse events [43].

Both anemia and malaria are common among pregnant women from sub-Saharan Africa. Goheen et al. conducted an in vitro observational cohort study involving an examination of *P. falciparum* pathogenesis in blood taken from anemic participants during their second and third trimesters [44]. RBCs were collected and assayed at baseline (*n* = 327), 14 days (*n* = 82), 49 days (*n* = 112), and 84 days (*n* = 115) after iron supplementation (60 mg ferrous fumarate daily). The researchers found that *P. falciparum* erythrocytic stage growth was reduced in blood from women at baseline but increased during supplementation. This paralleled increases in circulating CD71-positive reticulocytes and other markers of young RBCs. The conclusion was that *P. falciparum* growth in vitro is associated with elevated erythropoiesis, an essential step toward erythroid recovery in response to supplementation. These findings support WHO recommendations that iron supplementation be given in combination with malaria prevention and treatment services in malaria-endemic areas.

The treatment of IDA generally involves the oral administration of ferrous sulphate. Unfortunately, ferrous sulphate may have only a limited effect on hypoferremia and can cause adverse side effects. Though not strictly a micronutrient, iron-binding glycoprotein lactoferrin plays a central role in iron homeostasis and may offer therapeutic advantages both as a treatment for IDA in pregnancy and because of its bacteriostatic and bactericidal properties. This could be of value in reducing cervicovaginal infections, which may be one cause of preterm births. The mechanism whereby lactoferrin exerts its effect is hypothesized to be by reducing cytokine levels, notably IL-6, in cervicovaginal fluid. 

In a preliminary study, Giunta et al. evaluated the effectiveness of lactoferrin in preventing preterm delivery [45]. The study group consisted of 21 pregnant women (26–32 weeks’ gestation) with IDA who were considered at risk of preterm delivery (PTD). One group received twice daily oral 100 mg recombinant human lactoferrin (*n* = 14; lattoferrina; AG-pharma) and the other group (*n* = 7), 520 mg of ferrous sulphate (FerroGrad). The researchers found a correlation between one month of lactoferrin treatment in terms of a normalization of vaginal flora and a reduction in IL-6 in cervicovaginal fluid. Both groups had normal term births after 37 weeks. These findings are supported by Paesano et al., who showed that combined administration of oral and intravaginal bovine lactoferrin in a sub-cohort of pregnant women with threatened PTD decreased IL-6 in both serum and cervicovaginal fluids, cervicovaginal prostaglandin F2a, and suppressed uterine contractility [46].

The potential value of lactoferrin in the treatment of anemia and anemia of inflammation (AI) in mixed groups of pregnant and non-pregnant women (minor β-thalassemia, hereditary thrombophilia, AI, mixed group) was investigated by Lepanto et al. in an interventional study [47]. Bovine milk derivative lactoferrin (BL) has been shown to be a natural anti-inflammatory substance that can influence hepcidin and ferroportin synthesis. Compared with standard ferrous sulphate treatment (329.7 mg/day), 100 mg of 20–30% iron-saturated BL was found to be a more effective treatment for anemia and AI across all groups through its ability to downregulate IL-6.

### 3.7. Selenium

Selenium is an antioxidant that helps protect the body against the damaging effects of free radicals and cellular damage. Deficiencies of selenium are associated with accelerated disease progression and increased mortality among people infected with HIV. Similarly, low selenium levels in HIV-infected mothers increase the risk of preterm birth and the delivery of low-birth-weight neonates. Okunade et al. [48] undertook a study looking at the effects of selenium supplementation on pregnancy outcomes and disease progression in a Nigerian population of HIV-infected pregnant women. A total of 90 women were randomly assigned to receive either a daily 200 μg selenium or placebo tablet. The researchers found that women in the selenium arm had a significantly lower risk of preterm delivery (RR 0.32; 95% CI 0.11–0.96) and a nonsignificant reduction in the risk of delivering term neonates with a low birth weight (RR 0.24; 95% CI 0.05–1.19). Supplemental selenium did not increase the risk of perinatal death or adverse drug events. There were no significant effects on HIV disease progression. 

Thyroid dysfunction has been linked with a variety of maternal and fetal risks during pregnancy. The presence of thyroid autoantibodies in pregnant women has been associated with pregnancy complications such as miscarriage, preterm delivery, and postpartum thyroid dysfunction, even when levels of thyroid hormones are normal. In addition, adequate thyroid hormone levels are required for optimal development of the fetal brain and the nervous system [49]. Selenium may have a protective role. In the thyroid, the highest concentrations of selenium are encountered in form of selenoproteins, which are essential for thyroid hormone synthesis. Low levels of selenium are associated with thyroid disease, and there is evidence that selenium supplementation reduces the antibody titer in thyroiditis and may protect against thyroid autoimmunity during pregnancy [50].

In a placebo-controlled study by Mao et al., the effect of selenium supplementation on thyroid function in pregnant women with mild to moderate iodine deficiency was analyzed [51]. The impact of the supplementation on thyroid hormones differs depending on the presence or absence of thyroid antibodies. In women without autoantibodies, thyrotropin (TSH) increased and free thyroxin (FT4) decreased significantly during gestation (*p* < 0.001) in both treatment groups; these outcomes were previously reported as a normal adaptive processes in pregnancy. However, in the presence of autoantibodies, TSH and FT4 tended to decrease more in the selenium-treated group compared with placebo at 35 weeks (*p* ≤ 0.050). Regarding autoantibody levels, the authors found no statistical differences between the groups.

However, in another study by Mantovani et al., selenium supplementation had no effect on TSH or other parameters such as thyroid echogenicity, nor adverse pregnancy outcome [50]. Thyroglobulin antibodies (TgAb) and thyroid peroxidase autoantibodies (TPOAb) decreased significantly in both the selenium-treated and placebo groups during pregnancy; this might reflect the general modifications in maternal immune response. Importantly, selenium administration was found to reduce the levels of TgAB and TPOAb six months after delivery.

### 3.8. Zinc

Preterm premature rupture of membranes (PPROM) prior to 37 weeks’ gestation is associated with infant mortality and maternal and neonatal complications. History of PPROM is a risk factor for recurrence. It is thought that zinc has important effects on immunity and antioxidation by influencing the tensile strength of collagen in the fetal membranes. Furthermore, that disruption of collagen occurs as a result of the effects of zinc-dependent matrix metalloproteinase and tissue-specific inhibitors. The effect of supplementation (daily 40 mg zinc sulphate) vs. placebo treatment to prevent PPROM during the second and early third trimesters in pregnant women who had previously been affected was conducted by Shahnazi et al. [52] among women attending an Iranian midwifery clinic. Unfortunately, the researchers were unable to demonstrate any superiority of zinc supplementation on the prevention of PPROM and improvement of gestational age at birth or any anthropometric measurements. 

In a randomized, controlled trial from Indonesia, Helmizar examined the benefits of dadih, a dairy product from fermented buffalo milk, alone in or combination with zinc among pregnant women from West Sumatra [53]. A total of 138 pregnant women in their second trimester were randomly assigned to a control group, a dadih-only group (100 g six times per week) or a dadih-combination group (20 mg zinc/day plus 100 g six times per week) for six months. Dadih and zinc supplementation appeared to promote maternal and infant weight gain compared with the control group. By comparison, there was a nonsignificant reduction in maternal fecal secretory IgA (sIgA) between baseline and the endpoint in all groups, whereas infant sIgA significantly increased in the dadih and zinc group. The author concluded that supplementation improved pregnancy outcomes but had no effect on maternal immune response.

### 3.9. Omega-3 Fatty Acids

The effects of omega-3 fatty acids (FAs) on the immune system continues to confound researchers, particularly the value of the two main components of fish and algal oil, EPA and DHA.

Low maternal intake of omega-3 FAs has been linked with an increased risk of perinatal or postnatal depression. Omega-3, in particular EPA, appears to have an antidepressant action, possibly mediated via its anti-inflammatory actions. In a secondary analysis of a randomized, controlled trial, Mozurkewich et al. examined the effect of EPA- and DHA-rich fish oil supplementation on maternal and umbilical cord blood cytokines in pregnant women at risk of depression (Table 3) [54].

The investigators found that prenatal EPA-rich fish oil significantly reduced levels of IL6, IL15, and TNFα in maternal plasma, while prenatal DHA-rich fish oil had no significant effect on cytokine concentrations. By comparison, supplementation with EPA- and DHA-rich fish oil had no significant effect on umbilical cord blood cytokine concentrations [54].

In another study looking at depression in pregnancy, Nishi et al. investigated the link between symptoms, levels of estradiol (E2), inflammatory cytokines, and the benefits of EPA-rich omega-3 FA supplementation. E2 is an estrogen steroid hormone that increases during pregnancy and may modulate serotonin metabolism and reuptake transporters, and thus have an antidepressant effect [55]. Whether the antidepressant effects of EPA and E2 are cytokine-mediated is unclear.

Pregnant women at risk of depression were recruited at 12–24 weeks’ gestation and randomized to receive either 1800 mg omega-3 FA (9 capsules/day containing a total of 1206 mg EPA and 609 mg DHA) or placebo treatment for 12 weeks. E2, omega-3 FAs, CRP, interleukin-6, and adiponectin were measured at baseline and again after 12 weeks’ follow-up. Out of the 108 participants, 100 completed the study. Multivariable regression analyses demonstrated increased levels of EPA and E2 in association with a significant decrease in depressive symptoms in the omega-3 FA group, but not in the placebo group. Neither E2 nor omega-3 FA were associated with a change in inflammatory cytokines. The researchers concluded that supplementation with EPA and increased levels of E2 might function together to alleviate antenatal depression through a mechanism other than anti-inflammation [55].

Obese pregnant women have higher levels of insulin resistance compared with healthy weight women, and associated increases in inflammation in their adipose tissue. Haghiac et al. conducted a randomized, controlled trial that aimed to characterize the effects of omega-3 FA supplementation on inflammatory status in the placenta and adipose tissue of overweight and obese pregnant women [56]. After 25 weeks of supplementation (capsules containing EPA, 20:5n-3 plus DHA, 22:6n-3, total 2 g/day) or placebo from week 10–16 weeks’ gestation to term, investigators found that omega-3 FA decreased inflammation. This was demonstrated by lower expression of IL-6, IL-8, TNF-α, and TLR4 mRNA in the adipose tissue and placenta, and decreased plasma C-reactive proteins at the time of delivery. The results suggested that the decrease in inflammation was via the TLR4-induced innate immune response in adipose and trophoblast cells [56]. These findings support the hypothesis that omega-3 FA supplementation during pregnancy decreases obesity-associated tissue inflammation. However, future studies are still needed to define the precise mechanisms.

Increased levels of omega-3 FAs associated with dietary fish intake may have a preventative effect against the development of type-1 diabetes mellitus (T1DM). In a randomized, controlled trial by Horvaticek et al., the investigators demonstrated the positive effects of pregnancy and supplementation (capsules, total EPA 120 mg and DHA 616 mg/day) on C-peptide secretion in women with T1DM on a standard diabetic diet [57]. These findings suggest that omega-3 FA supplementation during pregnancy produces immunological tolerance and promotes production of endogenous insulin in women with T1DM.

By way of comparison with the above, various studies have failed to demonstrate an anti-inflammatory response or any benefit from dietary omega-3 FA supplementation during pregnancy. In a randomized, controlled trial, Forsberg et al. investigated the effect of supplementation with the probiotic *L. reuteri* oil drops, omega-3 FA (3 capsules containing 640 mg omega-3 FA), a combination, or placebo twice daily during pregnancy and lactation on maternal peripheral immunity [58]. The investigators found that after 20 weeks’ supplementation with *L. reuteri* during the second half of the pregnancy, levels of activated and resting regulatory T cells were lower in peripheral blood compared with the other groups. There was no reported effect of omega-3 FA supplementation on immune cell population in peripheral blood [58].

The potential association between preterm delivery, omega-3 FA supplementation, and a fish diet was investigated in a randomized trial in women with a history of preterm delivery. An ancillary study conducted by Harper et al. looked for changes in immune response [59]. The researchers found that maternal omega-3 FA supplementation or a fish diet did not affect preterm births. Furthermore, there was no significant difference in IL-10 and TNF-α levels between the omega-3 FA and placebo groups [59].

Keelan et al. conducted a randomized, controlled trial to determine whether fish oil-derived omega-3 FA supplementation during pregnancy from 20-weeks’ gestation altered placental polyunsaturated fatty acid composition, accumulation of specialized pro-resolving lipid mediators (SPMs), and inflammatory gene expression [60]. The investigators found that after -omega-3 FA supplementation, concentrations of the two main SPM precursors, 18-HEPE and 17-HDHA, were significantly increased in the placenta, as were placental DHA levels. Placental SPMs and EPA levels were not affected by omega-3 FA supplementation. Even though there was an increase in SPM precursor levels, there was no effect of PTGS2, IL-1β, IL-6, or IL-10 expression or a correlation between placental inflammatory genes observed [60].

### 3.10. Micronutrients, Malaria, and HIV

HIV and malaria infections commonly coexist, especially among pregnant women and young children in sub-Saharan Africa. The potential benefits of micronutrient supplementation on maternal immunity and susceptibility to malaria or disease symptoms have been touched on by several researchers (Table 4).

The lack of benefit from iron supplementation on *Plasmodium* infections has been discussed above [42]. Similarly, it is reported above that vitamin A and zinc have not proven of value on polymerase chain reaction–positive malaria, SGA, or prematurity [26]. By comparison, Goheen et al., in an in vitro study looking at the effects of iron on RBCs (see above) found that *P. falciparum* erythrocytic stage growth was reduced in women at baseline, but increased during supplementation [44]. This paralleled increases in circulating CD71-positive reticulocytes and other markers of young RBCs, findings that support WHO recommendations regarding iron supplementation in combination with malaria prevention and treatment services in endemic areas.

As part of an ongoing study, Chandrasiri et al. investigated the impact of lipid-based nutrient supplements on antimalarial antibodies in 1009 pregnant Malawian women before 20 weeks’ gestation and at 36 weeks’ gestation [61]. Participants received a daily lipid-based nutrient supplement (LNS), multiple micronutrients, or iron and folic acid. The source of the LNS is unclear from this paper, but may be based on Nutributter. The researchers found that antibodies to placental-binding isolates significantly increased, while antibodies to most merozoite antigens declined during pregnancy. The type of supplementation did not influence antibody levels at 36 weeks’ gestation or the rate of change in antibody levels from enrollment to 36 weeks’ gestation. Those with a higher BMI or coming from a higher socioeconomic class had significantly lower IgG and opsonizing antibodies to placental-binding antigens. 

Maternal infections are associated with maternal and fetal adverse outcomes. It is postulated that supplementation may reduce infections by improving maternal immunity. As part of the same Malawian study, Nkhoma et al. examined whether the above treatments influenced the occurrence of *P. falciparum* parasitemia during pregnancy, and trichomoniasis, vaginal candidiasis, and urinary tract infection (UTI) after delivery [62]. Again, there were no differences between intervention groups in the prevalence of any of the infections.

A randomized, placebo-controlled study by Olofin et al. explored the effects of daily multivitamin supplementation (vitamin B complex, C, and E) or vitamin A on malaria among pregnant women with HIV from Tanzania [28]. Women received malaria prophylaxis during their pregnancy and were reviewed monthly during the prenatal and postpartum periods. Malaria diagnosis was presumptive based on symptoms and/or by laboratory investigation. The researchers found that multivitamin supplementation lowered the risk of clinical malaria (RR 0.78; 95% CI 0.67–0.92) compared with receiving no multivitamins. Perhaps confusingly, multivitamins were found to increase the risk of any malaria parasitemia (RR 1.24; 95% CI 1.02–1.50). During this study, vitamin A supplementation did not change the malaria incidence. 

In Section 3.7, a study by Okunade et al. [48] is referred, which demonstrated in a Nigerian population of HIV-infected pregnant women that selenium lowered the risk of preterm delivery, but had no effects on disease progression.

### 3.11. Multiple Micronutrient Supplementation

The studies of Chandrasiri et al. [61] and Olofin et al. [28] involving multiple micronutrients in malaria are referred to above.

Finally, there were some interesting data on the effects of multiple micronutrients (MMN) based on a reanalysis of blood samples from 44 Indonesian maternal-child dyads participating in the Supplementation with Multiple Micronutrients Intervention Trial [63]. The researchers concluded that MMN supplementation affects maternal biomarker patterns of metabolism and inflammation in pregnancy, notably reducing CRP levels, and may have a similar effect in offspring. However, infant nutrition postpartum may have a greater impact on metabolism and inflammation.

## 4. Discussion

This scoping review covering an 11-year period found limited publications that directly explored maternal immunity in pregnancy and the effects of micronutrients. Often, the focus was elsewhere, with limited data regarding parameters of potential relevance to the mother. From what has been identified among a diverse collection of articles, it can be concluded that supplementation may contribute to some biochemical and clinical changes of potential immunological interest, but whether these affect maternal health or are widely applicable in justifying a change in supplementation guidelines is unknown.

### 4.1. Clinical Benefits of Supplementation

Were there demonstrable clinical benefits of micronutrient supplementation? The largest number of relevant publications demonstrating potential benefits were identified for vitamin D. Samimi et al. showed the beneficial effect of administrating vitamin D plus calcium during pregnancy on glycemic status, high-density lipoprotein cholesterol, plasma total glutathione concentration, and blood pressure [38]. The work of Jefferson et al. suggests that vitamin D supplementation may promote mucosal integrity and/or alters the vaginal environment with a possible effect on pregnancy outcomes [39]. 

Sun et al. investigated the value of iron combined with retinol supplementation in anemic pregnant women [27]. The researchers concluded that iron combined with retinol was more beneficial in improving iron status and lymphocyte proliferation during pregnancy than iron alone. Giunta et al. evaluated the effectiveness of lactoferrin in preventing preterm delivery. While all participants had normal term births after 37 weeks, the researchers demonstrated a potential correlation between one month of lactoferrin treatment and a normalization of vaginal flora and a reduction in IL-6 in cervicovaginal fluid [45].

Below, the work Nishi et al. is mentioned, who found that increased levels of EPA may reduce the risk of depressive symptoms during pregnancy [55]. Similarly, Horvaticek et al. suggest that omega-3 supplementation may promote production of endogenous insulin in pregnant women with T1DM [57].

### 4.2. Omega-3 Fatty Acids

Evidence regarding the potential benefits of omega-3 FAs s in pregnancy and the mechanism of action remain unresolved. High maternal consumption of fish oil-derived omega-3 may reduce complications associated with inflammation such as preterm delivery, preeclampsia, gestational diabetes, and perinatal depression [54]. A possible mechanism for these inflammation-modulating effects is postulated to be via alterations in cytokine production, notably IL-β, IL-6, and TNF-α. 

This scoping review has identified three studies looking at clinical outcomes [55,57,59] and four where the focus was on biochemical changes [54,56,58,60]. 

One of clinical relevance includes work by Nishi et al., who found that increased levels of EPA and E2 were associated with a significant decrease in depressive symptoms among at-risk pregnant women in the omega-3 FA group, but not in the placebo group. The researchers concluded that supplementation with EPA and increased levels of E2 might function together to alleviate antenatal depression through a mechanism other than anti-inflammation [55]. Similarly, findings from a randomized, controlled trial by Horvaticek et al., suggests that omega-3 FA supplementation during pregnancy produces immunological tolerance and promotes production of endogenous insulin in women with T1DM [57]. By comparison, a study conducted by Harper et al. looking for changes in immune response found that maternal omega-3 FA supplementation or a fish diet did not affect preterm births. Furthermore, there was no significant difference in IL-10 or TNF-α levels between the omega-3 FA and placebo groups [59]. 

Again, the results of biochemical studies paint a mixed picture. Mozurkewich et al. found that prenatal EPA-rich fish oil significantly reduced levels of IL6, IL15, and TNFα in maternal plasma, while prenatal DHA-rich fish oil had no significant effect on cytokine concentrations [54]. Haghiac et al. found that omega-3 FAs decreased inflammation in the placenta and adipose tissue of overweight or obese pregnant women [56]. Meanwhile, Forsberg et al. reported no effect of omega-3 FA supplementation on immune cell population in peripheral blood of pregnant women in comparison with supplementation with the probiotic *L. reuteri* [58]. Finally, Keelan et al. found that DHA and major specialized pro-resolving lipid mediators precursor levels were increased in the placenta of women who took omega-3 FA supplementation from 20 weeks’ gestation. The authors concluded that further studies are needed to ascertain pharmacological applications of omega-3 FA supplementation in pregnancy [60].

### 4.3. Genomic Effects

From the material above, it can be seen that the potential clinical benefits of vitamin D during pregnancy remain to be elucidated. However, supplementation does appear to be influencing the genomic landscape, which is already undergoing dramatic physiological adaptation as gestation progresses. 

Mirzakhani et al. performed transcriptome analysis as part of a nested case-control study of 157 women [32]. The researchers found that a subgroup of 47 participants who developed preeclampsia showed differential expression of 348 vitamin D–associated genes (158 upregulated) in their peripheral blood. 

Exploring these changes longitudinally between the first and third trimesters, Al-Garawi et al. confirmed alterations in maternal gene expression during pregnancy and that these changes were related to vitamin D levels [33]. Further gene expression and network analysis data indicated that the dysregulation of immune response pathways due to early pregnancy vitamin D insufficiency may contribute to the pathobiology of spontaneous preterm births [64].

In similar research on peripheral blood, Anderson et al. found that high-dose maternal vitamin D supplementation during pregnancy and lactation altered DNA methylation in mothers and breastfed infants compared with controls [34]. The associated gene clusters mapped to a range of activities (e.g., cell migration and motility, cell membrane processes, cadherin signaling, and immune function). What remains unclear is whether vitamin D directly influences gene regulation, and whether this impacts maternal health and fetal development. 

Choline may also alter the genomic landscape: while it is a constitutional compound of membranes in the form of phospholipids, highly enriched in all parenchymas, it is also a methyl donor for homocysteine conversion to methionine and DNA/histone methylation. In a controlled feeding study looking at the effects of two doses of choline in pregnant (26–29 weeks’ gestation) and non-pregnant women, Jiang et al. [41] demonstrated upregulation of the innate immune system among the former. This may help protect the mother and fetus against pathogens. The researchers found that healthy, third-trimester pregnant women experienced transcriptional activation of neutrophils in the peripheral innate immune system and elevated leukocyte oxidative DNA damage compared with non-pregnant women.

### 4.4. Timing of Administration

One reason for a lack of effect in many studies may be the timing of supplementation. It is known that fundamental changes are occurring within the womb and in the fetus during the first trimester of pregnancy. Khatiwada et al. looked at the longitudinal impact of administering either 400 or 4400 IU vitamin D3 from enrollment at 10–14 weeks’ gestation upon a range of parameters [37]. They found an association between baseline 25(OH)D and baseline TGF-β and second and third trimester IFN-γ and IL-2. Disappointingly, vitamin D supplementation throughout pregnancy did not impact immune mediators in later trimesters. The authors suggest that supplementing before conception could be key in influencing immune-mediator responses during pregnancy. 

This hypothesis is supported by the work by Mirzakhani et al., which examined the effects of early administration (from weeks 10 to 18) of either a standard recommended dose or high dose of vitamin D on changes in gene expression profiles and the occurrence of preeclampsia [32]. While there was no difference between treatment arms, the authors noted the emergence in early pregnancy of a distinctive immune response in the peripheral blood of 47 patients who developed preeclampsia. Interestingly, participants who already had 25-OH vitamin concentrations ≥30 ng/mL at trial entry and in late pregnancy appeared to be relatively protected.

### 4.5. Comparison with Animal Studies

Our diverse findings are disappointing when compared with the more definitive results obtained from animal studies demonstrating that micronutrient supplementation can affect biochemical and clinical aspects of immunity [8,9,10,11,12]. Chandra et al., for example, found that vitamin E and zinc supplementation improved the immune response during the peripartum period in Sahiwal cows [8]. It is, of course, recognized that animal studies are easier to conduct under controlled conditions and are less affected by ethical considerations and the need for informed consent.

### 4.6. Limitations of This Review

The present authors have concentrated on a limited search period, supplementation with micronutrients and omega-3 fatty acids and restricted themselves to the inclusion of only English articles in peer-reviewed journals. It is recognized that the inclusion of studies that examine the value of adding certain foods such as oily fish to the diet of pregnant women may have provided the reader with a more complete picture [65]. Also, there has been a wealth of studies looking at the value of probiotics in pregnancy that have not been referred to. A useful source of information here is the work by Lindsay et al. [66]. 

Many of the publications have examined methodological weaknesses or focus on limited parameters within a dynamic system. Some studies were potentially interesting but lacked detail (e.g., a paper on the potential benefits of dadih and zinc on pregnancy outcomes and humoral immune response) [53]. Others offered test results as evidence of a potential immunological effect. Priliani et al., for example, found that maternal biomarker patterns for metabolism and inflammation in pregnancy were influenced by multiple-micronutrient supplementation but failed to demonstrate a clinical benefit for the mother [63]. Whether the work of Ahmad et al. [25] or Siddiqua et al. [29] demonstrates an enhanced prenatal H1N1 vaccine response with vitamin A and vitamin B12 supplementation, respectively, is unknown.

## 5. Conclusions

This scoping review found limited publications that directly explored maternal immunity in pregnancy and the effects of micronutrients. From the identified data, it is concluded that supplementation may influence some clinical, biochemical, and genomic changes linked to maternal immunity. How generalizable and relevant these are to maternal health remains to be determined. A further problem when considering the relevance of any findings to current guidelines is that the latter vary internationally according to baseline and location.

In view of current research interest in the individual elements of this topic, it is surprising that so little attention has been devoted to a holistic examination of the role micronutrients may play in optimizing immunity during pregnancy. Rather than concentrating on measures that benefit the child, perhaps the starting point should be to promote the health of the mother. The German philosopher Ludwig Feuerbach said, “We are what we eat,” and if this is indeed the case, the value of micronutrients requires more attention [67].

Pregnancy is a complex process, and simply exploring a few short-term parameters in isolation is unlikely to provide an overview of a dynamic process. Additional work is needed to fully elucidate the biologic effects of micronutrient supplementation at varying doses and the timing/duration of their administration. Research findings could have implications for establishing clinical recommendations affecting prenatal and offspring health promotion.

## Figures and Tables

**Figure 1 nutrients-14-00367-f001:**
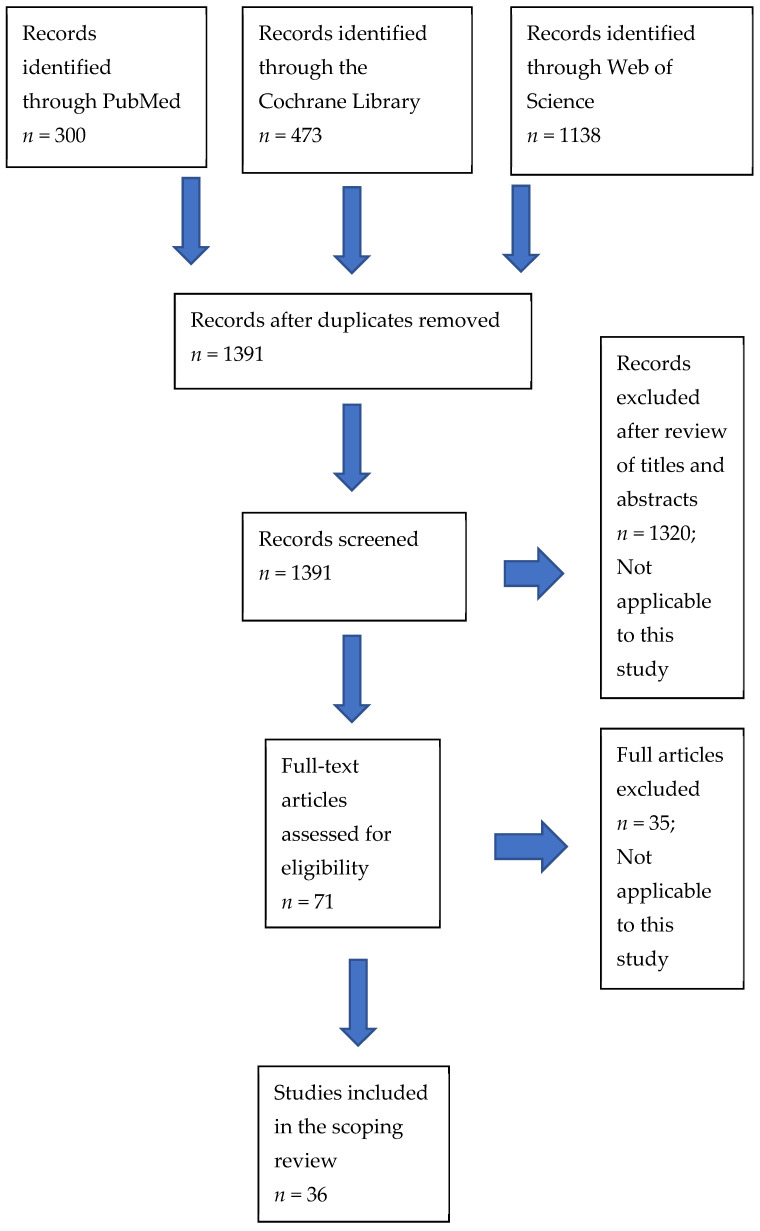
PRISMA diagram according to PRISMA-ScR guidelines [23].

**Table 1 nutrients-14-00367-t001:** Search terms.

Search Terms
(vitamin D OR vitamin D2 OR vitamin D3 OR cholecalciferol OR ergocalciferol OR calcitriol) AND immun* AND pregnan*
choline AND immun* AND pregnan* selenium AND immun* AND pregnan* zinc AND immun* AND pregnan* iron AND immun* AND pregnan* (vitamin B12 OR cobalamin) AND immun* AND pregnan* (vitamin A OR retinol OR beta carotene OR B carotene) AND immun* AND pregnan* (oily fish OR DHA OR docosahexaenoic acid OR EPA OR eicosapentaenoic acid) AND immun* AND pregnan* probiotic AND immun* AND pregnan* iodine AND immun* AND pregnan* (vitamin C OR ascorbic acid) AND immun* AND pregnan* obesity AND immun* AND pregnan* malnutrition AND immun* AND pregnan*

**Table 2 nutrients-14-00367-t002:** Effects of vitamin D supplementation.

Author (Year), Title, Journal	Study Design	Sample and Setting	Objectives and Methodology	Results	Implications
Mirzakhani et al.; (2016) Early pregnancy vitamin D status and risk of preeclampsia. *J. Clin. Investig.* [32]	Reanalysis of data from the randomized Vitamin D Antenatal Asthma Reduction Trial [VDAART]. Also, a nested case-control subgroup study to investigate peripheral blood gene expression profiles.	Multicenter US study. A total 881 women were randomized to receive vitamin D supplementation (4400 vs. 400 IU/day) initiated early in pregnancy (10–18 weeks). The ITT analysis comprised 4400 IU group: *n* = 408 (mean age 27.5 ± 5.4, BMI 28.9 ± 7.7). 400 IU group: *n* = 408 (mean age 27.2 ± 5.6, BMI 29.9 ± 7.6).	The study objectives were to examine the effects of standard- or high-dose supplementation on preeclampsia. An additional element was a nested Case-control study of 157 women looking at peripheral blood vitamin D-associated gene expression at 10 to 18 weeks. The latter including 47 participants who developed preeclampsia.	Outcome data were available for 816, with 67 (8.2%) developing preeclampsia. ITT analysis found no differences between treatment or controls in the incidence of preeclampsia (8.08 vs. 8.33%, respectively; RR 0.97; 95% CI 0.61–1.53). Vitamin D levels ≥30 ng/mL at trial entry and in late pregnancy were associated with a lower risk of preeclampsia (*p* = 0.04). Transcriptome analysis found differential expression of 348 vitamin D-associated genes (158 upregulated) in blood of women who developed preeclampsia (*p* < 0.05).	Vitamin D supplementation initiated in weeks 10–18 did not reduce preeclampsia. Higher doses may be of benefit. Differentially expressed vitamin D-associated transcriptomes point to a distinctive immune response during early pregnancy in women developing preeclampsia.
Al-Garawi et al.; (2016) The role of vitamin D in the transcriptional program of human pregnancy. *PLoS ONE* [33]	Nested cohort subgroup study of participants in VDAART (see above).	A total of 30 randomly selected, healthy women (mean age 25.2 ± 5.6 years, mean BMI 32.6 ± 8.8 kg/m^2^, 10–18 weeks’ gestation), participating a multicenter, randomized, controlled trial of vitamin D supplementation (400 vs. 4400 IU) in pregnancy.	The study objectives were to investigate how gene expression profiles change during pregnancy and the effects of vitamin D supplementation. RNA was isolated from blood samples at enrolment and during the third trimester (32–38 weeks). Differentially expressed genes were identified using significance of analysis of microarrays, and clustered using a weighted gene co-expression network analysis. Gene-set enrichment was performed to identify major biological pathways.	Comparison of profiles between the first and third trimesters identified 5839 significantly differentially expressed genes (FDR < 0.05; 57% down regulated vs. 42% upregulated). Weighted gene co-expression network analysis clustered these transcripts into 14 co-expression modules, of which 2 (green and salmon representing the 61 and 241 probes, respectively) showed significant correlation with maternal vitamin D levels. Pathway analysis demonstrated genes mapped to immune defense pathways and extracellular matrix reorganization, as well as genes enriched in notch signaling and transcription factor networks.	Gene expression profiles of pregnant women change during gestation. Maternal vitamin D levels may influence transcription. Alterations in the maternal transcriptome may contribute to fetal immune imprinting and reduce allergic sensitization in early life.
Anderson et al.; (2018) Effects of maternal vitamin D supplementation on the maternal and infant epigenome. *Breastfeed Med.* [34]	Double-blind, randomized, controlled pilot study.	Single-center Midwestern hospital obstetric practice, USA. Pregnant women recruited between 24- and 28-weeks’ gestation randomized to received vitamin D3 400 IU (*n* = 6; control group) or 3800 IU (*n* = 7; intervention group) daily through to 4–6 weeks postpartum.	The objective of this study was to quantify the effects of vitamin D3 supplementation on DNA methylation in pregnant and lactating women and their breastfed infants. Epigenome-wide DNA methylation was quantified in leukocytes collected from mothers at birth and mother-infant dyads at 4–6 weeks postpartum.	High-dose maternal vitamin D supplementation alters DNA methylation in mothers and breastfed infants compared with controls. There were variable gains and losses. At birth, intervention group mothers showed DNA methylation gains at the 76 cystine-guanine (CpG) dinucleotide and losses at 89 CpG dinucleotides. The associated gene clusters map to cell migration/motility and cellular membrane function. At postpartum, the strongest biological relevance in mothers was for cadherin signaling and immune function.	Vitamin D status may affect maternal and infant health during gestation and lactation by influencing the epigenomic landscape.
Rafiee et al.; (2015) Altered TH17/Treg ratio in recurrent miscarriage after treatment with paternal lymphocytes and vitamin D3: a double-blind placebo-controlled study. *Iran J. Immunol.* [35]	Double-blinded, placebo-controlled study.	Study conducted at Isfahan University of Medical Sciences, Iran between October 2013, and September 2014. A total of 44 patients with primary recurrent abortion (mean age 27.2 ± 5.0 years) were randomly assigned to the treatment (*n* = 22) and control (*n* = 22) groups.	The study objectives were to investigate the effects of vitamin D3 supplementation to enhance immune tolerance in women undergoing lymphocyte immune therapy (LIT) for recurrent miscarriages (RM). Of interest were the effect of vitamin D3 on the imbalance of 2 essential T cells subsets, Th17 and T regulatory (Treg) cells, in RM patients pre- and 3 months post treatment with LIT alone or in combination with vitamin D3.	Vitamin D3 therapy decreased the frequency of Th17 cells in addition to reducing the Th17/Treg ratio in peripheral blood of RM patients compared with the control group (*p* < 0.05).	RM patients have a higher Th17/Treg ratio in their peripheral blood. Vitamin D3 therapy decreased both the frequency of Th17 cells and the Th17/Treg ratio compared with the control group (*p* < 0.05) and may be a therapeutic candidate in RM.
Zerofsky et al.; (2016) Daily cholecalciferol supplementation during pregnancy alters markers of regulatory immunity, inflammation, and clinical outcomes in a randomized, controlled trial. *J. Nutr.* [36]	Double-blinded, randomized, controlled study (Kellogg Foundation grant)—see below.	A total of 57 pregnant women (mean age 29.6 ± 4.8 years, median BMI 25.1 kg/m^2^, IQR 21.3–29.5) at the University of California were randomized to receive either cholecalciferol (400 IU/day) or cholecalciferol 2000 IU/day from <20 weeks’ gestation to delivery.	The objectives of this study were to assess the effects of vitamin D supplementation during pregnancy on vitamin D status and markers of immune function linked with adverse outcomes.	Supplementation with a higher dose significantly increased vitamin D status during pregnancy (*p* < 0.0001). Women receiving 2000 IU/day had 36% more interleukin-10+ regulatory CD4+ T cells at 36 weeks than did those in the 400-IU/day group (*p* < 0.007).	Supplementation with 2000 IU/day is more effective at increasing vitamin D status in pregnant women than 400 IU/day and is associated with increased regulatory T cell immunity. This may prevent adverse outcomes caused by excess inflammation.
Khatiwada et al., (2021) Effects of vitamin D supplementation on circulating concentrations of growth factors and immune-mediators in healthy women during pregnancy. *Pediatr. Res.* [37]	Double-blinded, randomized trial at Medical University of South Carolina (Kellogg Foundation grant—see above).	Pregnant women enrolled at 10–14 weeks’ gestation were randomized to 400 or 4400 IU vitamin D3/day. Data on health, safety, circulating 25(OH)D and 9 immune mediators were collected in each trimester. 400 IU group: *n* = 107 (mean age 28.9 ± 5.29, BMI 29.5 ± 7.95 kg/m^2^). 4400 IU group: *n* = 110 (mean age 28.3 ± 4.69, BMI 28.5 ± 6.80 kg/m^2^).	The study objectives were to investigate the effects of plasma vitamin D metabolite 25(OH)D on plasma immune-mediators during the second and third trimesters of pregnancy, notably whether there exists an association between circulating blood levels and pro-inflammatory and tolerogenic immune-mediator concentrations.	Immune mediators in pregnant women were influenced by baseline (first trimester) plasma 25(OH)D values rather than increased levels secondary to supplementation. Baseline 25(OH)D was associated with baseline TGF-β and with IFN-γ and IL-2 in the second and third trimesters. Baseline IFN-γ, CRP, TGF-β, TNF-α, VEGF, IL-2, and IL-4 were associated with respective immune mediator concentrations in the second and third trimesters, notably at higher vitamin D3 dosage. Immune mediators were not affected by 25(OH)D concentrations in the second and third trimesters. Race was associated with baseline TGF-β, VEGF, and IL-10, and with IL-10 in the second and third trimesters.	Vitamin D supplementation before conception or during early in pregnancy, rather than later pregnancy, may be important when seeking to impact maternal immune-mediator response.
Samimi et al.; (2016) The effects of vitamin D plus calcium supplementation on metabolic profiles, biomarkers of inflammation, oxidative stress, and pregnancy outcomes in pregnant women at high risk of preeclampsia. *J. Hum. Nutr. Diet.* [38]	Prospective, double-blind, placebo-controlled trial.	A total of 60 primigravida women aged 18-40 years at risk for preeclampsia (as determined biochemical and on ultrasound scanning) attending the Kashan University of Medical Science, Iran. Participants were randomized to either 50,000 IU vitamin D3 every 2 weeks plus 1000 mg/day calcium carbonate supplement or to receive a placebo from 20 to 32 weeks’ gestation.	The study objectives were to examine the effects of vitamin D plus calcium administration on metabolic profiles and pregnancy outcomes among women at risk for pre-eclampsia. Treatment group *n* = 30 (27.3 ± 3.7 years, BMI 27.4 ± 3.3 kg/m^2^). Placebo group *n* = 30 (27.1 ± 5.2 years, BMI 25.6 ± 4.0 kg/m^2^). A range of biochemical and clinical parameters were investigated.	Taking both vitamin D3 and calcium supplements resulted in a significant reduction in fasting plasma glucose, serum insulin concentrations, insulin resistance, and beta cell function, and a significant rise in insulin sensitivity. Additionally, pregnant women receiving the combination demonstrated increased serum high-density lipoprotein (HDL)-cholesterol and plasma total glutathione concentrations (GSH).	Administration of vitamin D plus calcium for 12 weeks produced beneficial effects on glycemic status, HDL-cholesterol, GSH, and blood pressure among women at risk for preeclampsia.
Jefferson et al., (2019) Relationship between vitamin D status and the vaginal microbiome during pregnancy. *J. Perinatol.* [39]	Prospective randomized study.	A total of 402 healthy pregnant women attending the Medical University of South Carolina from 1 January 2013, to 30 April 2018 were enrolled. 387 were randomized to receive normal (400 IU/day, control group) or a high-dose vitamin D supplement (4400 IU/day; treatment group). In the control group (*n* = 191), 142 were followed to delivery. In the treatment group (*n* = 196), 155 were followed to delivery. There were 19 miscarriages and108 subjects dropped out.	The objective of this study was to investigate the association between vitamin D status and the vaginal microbiome in different ethnic American groups during pregnancy. Complete information was available on 236 participants (mean age 29 years, range 18–42, BMI 33.2 kg/m^2^). These comprised American Indians (*n* = 2, 1%), Black Americans (*n* = 83, 35%), Hispanics (*n*–61, 26%), and White women (*n* = 90 (38%). There were 112 women in the control group and 124 in the treatment group. Blood samples for 25(OH)D were taken monthly. Vaginal swabs were generally taken at 3 timepoints.	The vaginal microbiome was significantly affected by gestational age and ethnicity. The presence of *Megasphaera* bacteria correlated negatively (*p* = 0.0187) with 25(OH)D in Black women. Among controls, White women exhibited a positive correlation between plasma 25(OH)D and profuse *L. crispatus*.	There is an association between plasma 25(OH)D concentration and certain vaginal bacteria.

BMI—basal metabolic index; CD4+—cluster of differentiation 4; CRP—C-reactive protein; FDR—false discovery rate; GSH—plasma total glutathione concentration; HDL—high-density lipoprotein; IFN-γ—Interferon Gamma; IL-2/IL-4—Interleukin-2/-4; ITT—intention to treat; IQR—interquartile range; IU—international units; TGF-β—Transforming growth factor beta; TNF-α—tumor necrosis factor alpha; VDAART—Vitamin D Antenatal Asthma Reduction Trial; VEGF—Vascular endothelial growth factor; 25(OH)D—25-hydroxyvitamin D.

**Table 3 nutrients-14-00367-t003:** Summary of omega-3 fatty acid publications.

Author (Year), Title, Journal	Study Design	Sample and Setting	Objectives and Methodology	Results	Implications
Mozurkewich et al.; (2018) Effect of prenatal EPA and DHA on maternal and umbilical cord blood cytokines. *BMC Pregn. Childbirth* [54]	Secondary analysis of a prospective, double-blinded, randomized, controlled trial of fish oil supplementation during pregnancy for prevention of depressive symptoms.	Participants recruited from the antenatal clinics of the University of Michigan Medical Center and St. Joseph Mercy Health System, USA between October 2008 and May 2011. Pregnant women (12–20 weeks’ gestation) at risk of depression based on an Edinburgh Postnatal Scale Score between 9 and 19 or a history of depression who consumed ≤2 portions of fish per week.	The objectives of this study were to investigate the effects of prenatal EPA- and DHA-rich fish oil supplementation on 16 maternal and fetal cytokine production. Participants were assigned to receive daily EPA-rich fish oil (1060 mg EPA, 274 mg DHA), DHA-rich fish oil (900 mg DHA, 180 mg EPA), or placebo. Maternal blood samples were collected at enrollment and after 34–36 weeks’ gestation. Umbilical cord blood was collected at delivery. Follow-up was until 6 weeks post-partum.	Originally, 126 were enrolled and 118 completed the trial. A total of 113 samples were available for cytokine analyses after supplementation (minimum 14 weeks). EPA-rich fish oil supplementation decreased IL-6, IL-15, and TNF-α plasma concentrations compared with placebo, while DHA had no effect. There was no significant difference in IL-1β, IL-2, IL-5, IL-8, IL-12P70, IL-17, INF-γ, or MCP1 between groups. A total of 102 cord blood samples were analyzed. There were no significant differences in cord blood cytokines between groups.	Women with perinatal depressive symptoms related to inflammation may benefit from EPA-rich fish oil supplementation to reduce plasma concentrations of inflammatory cytokines. Future research in depression should focus on the loirole of EPA and clarify the relationship between inflammatory cytokines and depressive symptoms.
Nishi et al.; (2020) Plasma estradiol levels and antidepressant effects of omega-3 fatty acids in pregnant women. *Brian Behav. Immun.* [55]	Double-blinded, parallel-group, randomized, controlled trial (Synchronized Trial on Expectant Mothers with Depressive Symptoms by Omega-3 FAs [SYNCHRO])	Multicenter trial at Tokyo Medical University, University of Toyama, Chiba University, National Center of Neurology and Psychiatry, and National Center for Child Health and Development, Japan, and China Medical University, Taiwan. A total of 108 pregnant women at 12–24 weeks’ gestation with an Edinburgh Postnatal Depression Scale score ≥9 who eat ≤3 portions of fish/week	The aims of this study were to examine the association between increased estradiol (E2) levels, inflammatory cytokines, and depressive symptoms in pregnant women, and whether these were affected by omega-3 FA supplementation. Participants were randomized to receive omega-3 FA capsules (total 1206 mg EPA, 609 mg DHA/day, *n* = 49, mean age 32.8 ± 5.3 years) or placebo (*n* = 51, mean age 32.6 ± 5.3 years) for 12 weeks. Blood samples were taken at baseline and 12-week follow-up.	A total of 100 participants completed blood sampling. Increases in EPA and E2 were significantly associated with a decrease in depressive symptoms in the omega-3 FA group. Unexpectedly, the placebo group showed an association between increase in EPA and increase in depressive symptoms.	EPA supplementation and increased E2 levels during pregnancy may work synergistically to reduce depressive symptoms through a mechanism other than anti-inflammation. EPA may improve depression in non-pregnant people through an anti-inflammatory effect. Future studies should investigate how EPA supplementation and E2 levels in treating depression could be utilized.
Haghiac et al.; (2015) Dietary omega-3 fatty acid supplementation reduces inflammation in obese pregnant women: a randomized double-blind controlled clinical trial. *PLoS ONE* [56]	Randomized, double-masked, controlled trial	A total of 72 obese pregnant women at MetroHealth Medical Center, Cleveland, Ohio, USA, were randomized to receive omega-3 FA supplementation (capsules containing EPA, 20:5n-3 plus DHA, 22:6n-3; total 2 g) or placebo twice a day from week 10–16 weeks’ gestation to term.	The objectives of this study were to characterize the effects of omega-3 FA supplementation on inflammatory status in the placenta and adipose tissue of overweight/obese pregnant women and cultured adipose and trophoblast cells. Data was available on 25 omega-3 FA- treated participants (mean age 27 ± 5 years, BMI 33 ± 6) and 24 placebo participants (mean age 27 ± 5 years, BMI 32 ± 6).	After 25 weeks of supplementation, the adipose tissue and placenta showed lower expression of TLR4, IL-6, IL-8, and TNF-α, and there was decreased plasma CRP at delivery. Increased birth weights were observed in the omega-3 FA group.	Omega-3 FA supplementation decreased obesity-associated tissue inflammation in pregnancy. TLR4 appears to have a central role. Additional randomized, controlled, trials are needed to clarify the effect of omega-3 supplementation on birth weight.
Horvaticek et al.; (2017) Effect of eicosapentaenoic acid and docosahexaenoic acid supplementation on C-peptide preservation in pregnant women with type-1 diabetes: randomized placebo controlled clinical trial. *Eur. J. Clin. Nutr.* [57]	Prospective, randomized, placebo-controlled clinical trial.	Conducted at Ministry of Health Referral Center for Diabetes in Pregnancy, Department of Obstetrics and Gynecology, Zagreb University Hospital, Republic of Croatia, 90 pregnant women with type 1 diabetes mellitus (T1DM between 5 and 30 years) at 9 weeks’ gestation were randomized to a standard diabetic diet plus EPA and DHA capsules (total 120 mg EPA, 616 mg DHA/day) or standard diabetic diet plus placebo.	The objectives of this study were to explore the effects of EPA and DHA supplementation on fasting C-(FC) peptide secretion in pregnant women with T1DM. A total of 47 women were randomized to EPA and DHA (mean age 29.8 ± 5.5 years, BMI 23.3 ± 3.3 kg/m^2^) and 43 to the placebo group (mean age 29.6 ± 4.8 years, BMI 22.9 ± 3.1 kg/m^2^). Blood samples for FC, fasting blood glucose (FBG), and HbA1c were analyzed during each trimester.	Supplementation with EPA and DHA resulted in a significant increase in FC-peptide during pregnancy. In the placebo group, the rise in FC-peptide was not significant. There were no differences in birth weight and prevalence of fetal macrosomia. C-peptide and FBG concentrations in umbilical vein serum were lower in the treatment group.	EPA and DHA supplementation in pregnancy cause immunological tolerance and stimulate the production of endogenous insulin in women with T1DM.
Forsberg et al.; (2020) Changes in peripheral immune populations during pregnancy and modulation by probiotics and ω-3 fatty acids. *Sci. Rep.* [58]	Prospective, randomized, double-blinded, placebo-controlled, allergy prevention trial (PROOM-3).	Multicenter trial conducted at the Department of Pediatrics and Allergy Center at University Hospital in Linköping and 3 county hospitals in Sweden. A total of 88 pregnant women at 20 weeks’ gestation with clinical symptoms/history of allergic disease, or a child/partner with clinical symptoms/history, were randomized into four groups to receive either *L. reuteri* oil drops, omega-3 FA (3 capsules containing 640 mg omega-3 FA), a combination, or placebo, twice daily during pregnancy and lactation.	Objectives of this study were to investigated how maternal peripheral immunity is affected by pregnancy and *L. reuteri* (probiotic) and omega-3 FA supplementation. The reseachers used flow cytometry and a broad panel of immune markers to map peripheral immune cell populations. Groups: *L. reuteri* oil drops (*n* = 23, mean age at inclusion 29 years, study product 18.8 weeks). omega-3 FA (*n* = 21, mean age 30 years, study product 18.6 weeks). *L. reuteri* + omega-3 FA (*n* = 22; mean age 29 years, study product 19.6 week). Placebo (*n* = 22, mean age 29 years, study product 18.5 week).	*L. reuteri* supplementation from 20 weeks’ gestation decreased activated and resting Treg cells at 4 days post-delivery compared with omega-3 FA and placebo. There were no significant differences between results for the omega-3 FA and placebo groups. Lymphocyte and monocyte populations were not affected by supplementation.	A total of 20 weeks of supplementation with *L. reuteri* during pregnancy resulted in immunomodulatory effects on activated and resting Treg cells. omega-3 FA supplementation had no effect in this study.
Harper et al.; (2013) Change in mononuclear leukocyte responsiveness in midpregnancy and subsequent preterm birth. *Obstet. Gynecol.* [59]	Ancillary study to a randomized, controlled trial of omega-3 FA supplementation to prevent recurrent preterm birth.	The cohort consisted of 852 women (mean age 27 ± 23–32 years) attending a US Eunice Kennedy Shriver National Institute of Child Health and Human Development Maternal-Fetal Medicine Unit who participated in the randomized trial. Pregnant women between 16- and 21-weeks’ gestation with prior spontaneous preterm births were randomized to receive omega-3 FA supplementation (2000 mg) or placebo.	The objectives of this study were to explore changes in immune response associated with preterm birth, omega-3 FA supplementation, and a fish diet history. Blood samples were taken at baseline (16–22 weeks) and again at 25–28 weeks’ gestation (follow-up) to examine in vitro maternal peripheral blood mononuclear leukocyte production of interleukin-10 (IL-10), tumor necrosis factor-α (TNF-α), in response to stimulation with lipopolysaccharid.	A total of 343 of 852 participants had paired cytokine measurements for either IL-10, TNF-α, or both. Anti-inflammatory IL-10 and proinflammatory TNF-α levels were unaffected by omega-3 FA supplementation or a fish diet. The rate of preterm birth at less than 37 weeks’ gestation was 33.7% for those who ate at least 1 portion of fish/week and 44.4% for those who ate <1 portion of fish/week.	Recurrent preterm birth before 35 weeks was associated with decreased peripheral blood mononuclear leukocyte production of IL-10 in response to a lipopolysaccharide stimulation during the second trimester. This study demonstrated that the variable influences of a fish diet and omega-3 FA supplementation on preterm birth rates were not due to a modulation effect on the maternal immune response.
Keelan et al.; (2015) Effects of maternal n-3 fatty acid supplementation on placental cytokines, pro-resolving lipid mediators, and their precursors. *Reproduction* [60]	Placentas were collected from women enrolled in a randomized, placebo-controlled trial of omega-3 FA supplementation from 20 weeks’ gestation.	Conducted at St John of God Hospital and Princess Margaret Hospital, Australia, between 1999 and 2001. A total of 98 pregnant women who ate ≤ than 2 portions of fish/week were randomized to receive omega-3 FA supplementation (3.7 g/day, 56% DHA and 27.7% EPA) or placebo.	The objectives of this study were to examine whether levels of specialized pro-resolving lipid mediators (SPMs) and their precursors varied in placental tissue from women taking omega-3 FA supplementation during pregnancy compared to a control population.	A total of 51 placentas were sampled. Omega-3 FA supplementation increased placental DHA and levels of SPM precursors, but not EPA. Expression of PTGS2, IL-1β, IL-6, and IL-10 was unaffected by omega-3 FA. Conversely, supplementation increased expression of TNF-α 14-fold.	DHA and major SPM precursors levels were increased in the placenta of women who took omega-3 FA supplementation from 20 weeks’ gestation.

BMI—basal metabolic index; CRP—C-reactive protein; DHA—docosahexaenoic acid; E2—estradiol 2; EPA—eicosapentaenoic acid; FBG—fasting blood glucose; FC—fasting C peptide; HbA1c –glycated hemoglobin; IL-1β/-2/-5/-6/-8/-12P70/-15/-17-Interleukin—1B/-2/-5/-6/-8/-12P70/-17; MCP1—monocyte chemoattractant protein-1; PTGS2—prostaglandin-endoperoxide synthase 2; SPM—specialized pro-resolving mediators; TLR4—toll-like receptor 4; TNF-α—tumor necrosis factor alpha; omega-3 FA—omega-3 fatty acids.

**Table 4 nutrients-14-00367-t004:** Micronutrients, malaria, and HIV.

Author (Year), Title, Journal	Study Design	Sample and Setting	Objectives and Methodology	Results	Implications
Mwangi et al.; (2015) Effect of daily antenatal iron supplementation on *Plasmodium* infection in Kenyan women: a randomized clinical trial. *JAMA* [42]	Randomized clinical trial.	The study involved a total of 470 pregnant rural Kenyan women. Participants were aged 15–45 years with singleton pregnancies and a gestational age of 13–23 weeks and hemoglobin concentration of ≥90 g/L. All women received 5.7 mg iron/day as fortified flour and the usual intermittent malaria preventive throughout the study with sulfadoxine-pyrimethamine.	The objectives of this study were to measure the effect of antenatal iron supplementation on maternal *P. falciparum* infection risk at birth, iron status, and neonatal outcomes during a malaria epidemic. The intervention group received daily supplementation with 60 mg iron (ferrous fumarate, *n* = 237; median age 24.0 years, IQR 20.0–28.5, BMI 22.1 ± 2.7 kg/m^2^) or placebo (*n* = 233, median age 24.0 years, IQR 20.0–29.0, BMI 21.8 ± 2.6 kg/m^2^)) from randomization until the first month postpartum.	A total of 40 women were lost to follow-up/excluded. At baseline, 190 of 318 women (59.7%) were iron-deficient. An ITT analysis comparing iron vs. placebo demonstrated a *P. falciparum* infection risk at birth of 50.9 vs. 52.1% (crude difference, −1.2%; 95% CI −11.8% to 9.5%; *p* = 0.83) and birth weight of 3202 vs. 3053 g, respectively. (crude difference 150 g; 95% CI 56–244; *p* = 0.002), respectively.	There were no differences in overall maternal *P. falciparum* infection risk between women taking iron supplementation or placebo. Iron supplementation led to increased birthweight. There was no evidence that iron supplementation caused serious adverse events.
Darling et al.; (2017) Vitamin A and zinc supplementation among pregnant women to prevent placental malaria: a randomized, double-blind, placebo-controlled trial in Tanzania *Am. J. Trop. Med. Hyg.* [26]	Randomized, double-blinded, placebo-controlled trial with a factorial design.	A total of 2500 HIV-negative primigravid or secundigravid pregnant women in their first trimester in Dar es Salaam, Tanzania.	The objectives of this study were to investigate whether supplementation with vitamin A, zinc, or both starting in the first trimester reduces rick of placental malaria and adverse pregnancy outcomes. A total of 625 participants were allocated to each treatment group: 2500 IU of vitamin A, 25 mg of zinc, both 2500 IU of vitamin A and 25 mg of zinc, or a placebo until delivery. Secondary outcomes included small for gestational age (SGA) births and prematurity. Follow-up was for at least 6 weeks post-delivery.	Placental samples were obtained in 1404 mothers (mean age 22.9 ± 4.4 years, BMI 23.5 ± 4.4 kg/m^2^), comprising 56% of participants and 62% of all pregnancies ≥28 weeks [*n* = 2266]). Birth outcomes were obtained for 2434 of 2500 randomized participants. Women who received zinc had a lower risk of histopathology placental malaria compared with those not receiving zinc (RR 0.64; 95% CI 0.44–0.91). PCR-positive malaria, SGA, and prematurity were not affected in either treatment group.	Pregnant women who received zinc had a lower risk of histopathology-positive placental malaria. None of the active treatments influenced PCR-positive malaria, SGA, or prematurity. There were no safety concerns.
Goheen et al.; (2017) Host iron status and erythropoietic response to iron supplementation determines susceptibility to the RBC stage of falciparum malaria during pregnancy. *Sci. Rep.* [44]	Observational cohort study.	Pregnant women (18–45 years old between 14 and 22 weeks’ gestation) from the Kiang West and Jarra East regions of rural Gambia. Participants were recruited between June 2014 and March 2016 from the reference arm of a randomized trial testing the efficacy and safety of a hepcidin-guided screen-and-treat strategy for combatting anemia.	The objective of this study was to investigate whether iron supplementation increased the risk of *P. falciparum* by means of an in-vitro study of RBCs from pregnant women during their second and third trimesters. RBCs were collected and assayed before (*n* = 327) and 14 days (*n* = 82), 49 days (*n* = 112), and 84 days (*n* = 115) after iron supplementation (60 mg iron as ferrous fumarate daily).	*P. falciparum* erythrocytic stage growth in vitro is reduced in anemic pregnant women at baseline, but increased during supplementation. The elevated growth rates paralleled increases in circulating CD71-positive reticulocytes and other markers of young RBCs.	In vitro, *P. falciparum* growth in response to iron supplementation is associated with elevated erythropoiesis, an essential step in erythroid recovery. These results support WHO recommendations regarding iron supplementation.
Chandrasiri et al.; (2015) The impact of lipid-based nutrient supplementation on anti-malarial antibodies in pregnant women in a randomized controlled trial. *Malar. J.* [61]	Single-blinded, randomized, controlled trial.	A total of 1009 pregnant Malawian women (median age 24, IQR 20–28, mean gestation 16.5 ± 2.20 weeks; median BMI 21.6, IQR 20.3–23.5 kg/m^2^) enrolled in the high-energy, micronutrient fortified lipid-based nutrient supplements (iLiNS-DYAD) trial. The source/composition of the lipid-based nutrient (LNS) is unclear. All participants received two doses of sulphadoxine-pyrimethamine (SP) malaria intermittent preventative treatment at enrolment and at 28–34 gestation weeks.	The objective of this study was to investigate whether different nutrient supplements offered to pregnant women reduced their susceptibility to malaria by improving immunity as judged by changes in antibody levels. Antibodies to antigens expressed by a placental-binding parasite isolate, a non-placental binding parasite isolate, merozoites, and schizonts were measured at enrollment (before 20 gestation weeks) and at 36 weeks in women receiving a daily lipid-based nutrient supplement, multiple micronutrients, or iron and folic acid.	Antibodies to placental-binding isolates significantly increased while antibodies to most merozoite antigens declined between timepoints. The type of supplementation did not affect antibody levels at 36 weeks’ gestation or their rate of change. There was a negative association between maternal BMI and antibodies to placental-binding antigens (coefficient [95% CI] −1.04 [−1.84, −0.24]).	Nutrient supplementation did not affect antimalarial antibody responses. Women with higher socioeconomic status had significantly lower IgG and opsonizing antibodies to placental-binding antigens that were not influenced by supplementation type.
Nkoma et al.; (2017) Providing lipid-based nutrient supplement during pregnancy does not reduce the risk of maternal P falciparum parasitemia and reproductive tract infections: a randomised controlled trial. *BMC Pregnancy Childbirth* [62]	Randomized, controlled trial.	Substudy of 1391 pregnant Malawian women enrolled in the iLiNS-DYAD trial between 2011 and 2013. See Chandrasiri et al. [61] above. The source/composition of the LNS is unclear.	The objectives of this study were to investigate the impact of daily lipid-based nutrient SQ-LNS (*n* = 462, mean age 25 ± 6, BMI 22.2 ± 3.0 kg/m^2^), multiple micronutrients (MMN, *n* = 466 mean age 25 ± 6 years, BMI 22.2 ± 2.9 kg/m^2^), or iron and folic acid (*n* = 463, mean age 25 ± 6, BMI 22.1 ± 2.6 kg/m^2^) from <20 weeks’ gestation on occurrence of *P. falciparum* parasitemia during pregnancy, and trichomoniasis, vaginal candidiasis, and UTI after delivery. Assessment was at 32 (RDT) and 36 weeks’ gestation (PCR), at delivery (RDT and PCR), and 1 week post-delivery (microscopy and urine analysis).	The prevalence of *P. falciparum* parasitemia was 10.7% at 32 weeks’ gestation, 9% at 36 weeks’ gestation, and 8.3% at delivery. The value for PCR testing at delivery was 20.2%. After delivery, the prevalence of trichomoniasis was 10.5%, vaginal candidiasis 0.5%, and UTI 3.1%. There were no differences between intervention groups in the prevalence of these infections.	SQ-LNS did not influence the occurrence of maternal *P. falciparum* parasitemia, trichomoniasis, vaginal candidiasis, or UTI.
Olofin et al.; (2014) Supplementation with multivitamins and vitamin A and incidence of malaria among HIV-infected Tanzanian women. *J. Acquir. Immune. Defic. Syndr.* [28]	Randomized, controlled trial with modifications.	A total of 1078 HIV-infected pregnant women (mostly second trimester) in Dar es Salaam, Tanzania, from April 1995 until August 2003, with modifications in 1998 and 2000. Malaria was defined by a presumptive clinical diagnosis and/or examination of blood smears for the malaria parasite. Women received malaria prophylaxis during pregnancy. Participants were followed up monthly during the prenatal and postpartum periods.	The object was to examine whether daily multivitamin supplementation (vitamin B complex, C, and E) or vitamin A supplementation altered malaria incidence in HIV-infected pregnant women. Four groups: multivitamins (20 mg vitamins B1, 20 mg B2, 25 mg B6, 100 mg niacin, 50 mg B12, 500 mg C, 30 mg E, and 800 mg folic acid), vitamin A alone (30 mg b-carotene with 5000 IU preformed vitamin A), both multivitamins and vitamin A, or placebo.	Median follow-up was 41.0 months (or to next pregnancy, loss to follow-up, or death). Multivitamin supplementation compared with no multivitamin ± s significantly lowered the risk of clinically diagnosed clinical malaria (RR 0.78; 95% CI 0.67–0.92). Vitamin A supplementation did not change malaria incidence. Multivitamins increased the risk of any malaria parasitemia (RR 1.24; 95% CI 1.02–1.50).	Multivitamin supplements protected against development of symptomatic malaria among pregnant, HIV-positive women. The clinical significance of increased malaria parasitemia among supplemented women is unknown.

BMI—basal metabolic index; CD71—cluster of differentiation 71; HIV—human immune deficiency virus; IgG—immunoglobulin G; iLiNS-DYAD—international lipid-based nutrient supplements project; ITT—intention to treat; IQR—interquartile range; IU—international units; LNS—lipid-based nutrient supplement; MMN—multiple micronutrients; PCR—polymerase chain reaction; RBC—red blood cell; RDT—rapid diagnostic testing; SGA—small for gestational age; UTI—urinary tract infection.

## Data Availability

All data used in this scoping review is publicly available.

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
