# Peer review of "Do Micronutrient and Omega-3 Fatty Acid Supplements Affect Human Maternal Immunity during Pregnancy? A Scoping Review"

_nutrients, 2022, doi:10.3390/nu14020367_

Round 1
Reviewer 1 Report
This review is very interesting, and the authors have the full support with respect to their statement in the Conclusions (l778-780): In view of current research interest in the individual elements of this topic, >>it is surprising that so little attention has been devoted to a holistic examination<< of the role micronutrients may play in optimizing immunity during pregnancy. Nevertheless, there are several shortcomings in this review:
Major:
- Abstract: It is to be clearer justified, why only 2.66% of existing leterature was included. This may exclude the validity of current knowledge.
- The conclusie statement of absent holistic approaches must clearly be pointed out in the abstract and intrduction.
- l. 118: Exlusion of fortification and particular food rather than supplementation is critical and must be justified/explained.
- l. 123: Was there no chance on including French and Russian trials, mostly having English abstracts?
- The processes into which special micronutrients are involved should be described in more detail, particularly their context with other potenttially critical micronutrients. This would have a great impact on the mostly insufficient design of studies where only isolated nutrients or a limited number of critical nutrients were given. E. g., it doesn't make sense to compare B-vitamins with vit. A. The reader should be able to capture the expectable failure of a trial, when Liebig's Law of the Minimum is neglected.
- l. 557: Generally, please define trial differences in terms of dosage, duration etc. Success of supplementation, e. g. with respect of preterm delivery, largely depends on the amount of omega-3-fatty acids administered, and which of them was given. Please do not simply use the term omega-3-fatty acid when describing a trial! Plant ALA, predominantyl EPA or predominantly DHA. If no basal data are avialable, this must be mentioned. Scandinavians and people from Texas have different basal values, and preterm rates as well!
- Iron and malaria/Table 4 Mwangi et al: Please specify the basal antimalarial treatment (includig the use of ubiquitous herbal antimalarial medicine (A. annua) if possible) to discussthe risk of study participants to increase plasmodial load in response to iron treatment.
- Table 4, Chandrasiri et al, Nkoma et al and l. 596: Please specify the term lipid-based nutrient supplementation.
- l. 720: Choline is a constitutional compound of membranes in the form of phospholipids, highly enriched in all parenchymas, not only a methyl donor for homocysteine conversion to methionine and DNA/histone methylation.
Minor:
- line 232f: Please specify! Do the authors mean calcitriol synthesis in ovary etc?
- Tables are hard stuff to read. Please use catch phrases to shorten text.
- l. 520: This is to be expected according to placental bioattenuation/biomagnification capacity.
- l. 551: development >of< type-1 diabetes
- l. 669: The reviewer would like to suggest to replace >potential< by >demonstrated<.
Author Response
- Abstract: It is to be clearer justified, why only 2.66% of existing leterature was included. This may exclude the validity of current knowledge. RESPONSE: Your point is well made. We have concentrated on only the recent literature (~12 years). at Line 99 we state: "The search period was chosen following a preliminary exploration of the topic and in anticipation that this would result in a manageable dataset that is likely to encompass the most relevant publications." Should we have missed any important publications prior to 2010, these are likely to have been mentioned among our selected articles.
- The conclusie statement of absent holistic approaches must clearly be pointed out in the abstract and intrduction. RESPONSE: Another important point. We have added text at Line 25 to say that "None provided a holistic perspective." We have also added similar text to Line 78 in the Introduction
- l. 118: Exlusion of fortification and particular food rather than supplementation is critical and must be justified/explained. RESPONSE: Our title refers to 'Supplementation'. Again, we had to adopt exclusion criteria which would make this scoping review manageable. We have added the following at Line 120: "Foods promoted for well-being typically contain a range of nutritional components making the identification of benefits associated with individual micronutrients problematic. Similarly, it is difficult to separate the effects of fortification from the elements of the product being added to."
- l. 123: Was there no chance on including French and Russian trials, mostly having English abstracts? RESPONSE: See our response to 1. above. Undoubtedly, there are excellent French and Russian studies which we have been unable to include in view of the magnitude of the topic. We are already at around 15,000 words. Also, we felt it important not just to work from abstracts or conference reports but to examine the whole research text.
- The processes into which special micronutrients are involved should be described in more detail, particularly their context with other potenttially critical micronutrients. This would have a great impact on the mostly insufficient design of studies where only isolated nutrients or a limited number of critical nutrients were given. E. g., it doesn't make sense to compare B-vitamins with vit. A. The reader should be able to capture the expectable failure of a trial, when Liebig's Law of the Minimum is neglected. RESPONSE: Thank you for reminding us of Liebig's Law which states; "Growth is dictated not by total resources available, but by the scarcest resource." Our aim with this initial scoping review was primarily to see what is out which may be of relevance to the mother, rather than exploring any comparable benefits.
- l. 557: Generally, please define trial differences in terms of dosage, duration etc. Success of supplementation, e. g. with respect of preterm delivery, largely depends on the amount of omega-3-fatty acids administered, and which of them was given. Please do not simply use the term omega-3-fatty acid when describing a trial! Plant ALA, predominantyl EPA or predominantly DHA. If no basal data are avialable, this must be mentioned. Scandinavians and people from Texas have different basal values, and preterm rates as well! RESPONSE:An important point. We have amended this section and associated table.
- Iron and malaria/Table 4 Mwangi et al: Please specify the basal antimalarial treatment (includig the use of ubiquitous herbal antimalarial medicine (A. annua) if possible) to discussthe risk of study participants to increase plasmodial load in response to iron treatment. RESPONSE: The usual intermittent malaria preventive throughout the study was with sulfadoxine-pyrimethamine. No mention is made of herbal treatments. The authors write that "Per-protocol analysis suggested that IPT use may have modified the effect of iron on hemoglobin concentration, but the absence of a clear trend with dose indicates that this finding may be spurious. Interpretation of these data is further complicated because IPT use was not a baseline factor and may have acted as a mediating factor." We decided not to explore this further in our scoping review.
- Table 4, Chandrasiri et al, Nkoma et al and l. 596: Please specify the term lipid-based nutrient supplementation. RESPONSE: No further details are to be found in this paper. Support from the International Lipid-based Nutrient Supplementation (iLiNS)-Project Steering Committee is acknowledged which appears to be linked to the Bill & Melinda Gates Foundation. In a possibly related trial protocol, we learn that "Lipid-based nutrient supplements (LNS) made using vegetable oil, groundnut paste, milk, sugar, and micronutrients." In the original Clinicaltrails.gov protocol for
Supplementing Maternal and Infant Diet With High-energy, Micronutrient Fortified Lipid-based Nutrient Supplements (LNS) (iLiNS-DYAD-M)
Nutributter and fortified spread as potential sources are mentioned. We have amended the text for both Chandrasiri and NKoma - l. 720: Choline is a constitutional compound of membranes in the form of phospholipids, highly enriched in all parenchymas, not only a methyl donor for homocysteine conversion to methionine and DNA/histone methylation. RESPONSE: We have reused your wording - thank you.
Minor:
- line 232f: Please specify! Do the authors mean calcitriol synthesis in ovary etc? RESPONSE: Yes, amended
- Tables are hard stuff to read. Please use catch phrases to shorten text. RESPONSE: we will work on shortening the tables
- l. 520: This is to be expected according to placental bioattenuation/biomagnification capacity.
- l. 551: development >of< type-1 diabetes RESPONSE: corrected
- l. 669: The reviewer would like to suggest to replace >potential< by >demonstrated<.RESPOSE: Modified
Reviewer 2 Report
STRUCTURE
- The manuscript is properly structured
TITLE AND ABSTRACT
- The Abstract should contain all the micronutrients analyzed (vitamin E is missing).
- Do not use the first person in scientific publications. Applicable to the whole document.
INTRODUCTION
- The introduction is brief, it should be justified with more literature and current scientific background on the subject, especially on other studies in which other micronutrients have been studied, such as selenium, vitamin B12, choline and probiotics.
- For example explain in detail the next statement that appears in the results which can be dealt with in depth in this section: "By comparison, at postpartum, maternal meth- 289 ylation gains, which are biologically relevant for cadherin signaling and immune function, were seen at 200 and a loss at 102 CpGs".
MATERIAL AND METHODS
- Divide the methodology into the different sections for a better understanding and approach (eligibility criteria, information sources, risk of bias and synthesis of results). Improve the structure, e.g., include data collection and synthesis and methodological quality in a different section.
- Present key elements of the study design at the beginning of the paper. For example: this literature review was conducted using the informative guidelines for scoping reviews [23]. It was designed following the recommendations of the PRISMA Statement… [CITE].
Tricco AC, Lillie E, Zarin W, O'Brien KK, Colquhoun H, Levac D, Moher D, Peters MDJ, Horsley T, Weeks L, Hempel S, Akl EA, Chang C, McGowan J, Stewart L, Hartling L, Aldcroft A, Wilson MG, Garritty C, Lewin S, Godfrey CM, Macdonald MT, Langlois EV, Soares-Weiser K, Moriarty J, Clifford T, Tunçalp Ö, Straus SE. PRISMA Extension for Scoping Reviews (PRISMA-ScR): Checklist and Explanation. Ann Intern Med. 2018 Oct 2;169(7):467-473. doi: 10.7326/M18-0850. Epub 2018 Sep 4. PMID: 30178033.
Bias
- It is recommended to describe any efforts to address potential sources of bias. Specify the methods used to assess risk of bias in the included studies.
RESULTS
- Give the total number of included studies and participants and summarise relevant characteristics of studies. There is no table with the main characteristics of the sample size (age, week of gestation, BMI, follow-up period, etc). It is proposed, for example, to create a study description section and include it.
- Create a study selection section and improve the PRISMA diagram with the reference indicated in the methodology section.
- Line 181, this section is focused in Vitamina B12 and it is discussed vitamin A?
- The information regarding studies [47] and [26] is repeated both in the selenium and HIV section and in the vitamin A and malaria section. The information should only appear once.
- When a table is split into two sheets, the sections must be put back in the header.
- Various abbreviations are used in the tables which are not explained. They should be clarified in the table footnotes.
- Include in the tables what the study objective was.
- Add all important harms or unintended effects for each study and micronutrient.
DISCUSSION
- Provide a brief summary of the limitations of the evidence included in the review (e.g. study risk of bias, inconsistency and imprecision).
- Provide a cautious overall interpretation of the results taking into account the objectives, limitations, multiplicity of analyses, results of similar studies, and other relevant evidence.
- Discuss in detail this statement: "Differences in DNA methylation were also seen 291 in breastfed infants of intervention mothers at 4-6 weeks". Compare DNA methylation with other studies.
- Discuss the generalizability (external validity) of each of the sections included in the study.
- Compare the results obtained with current practice guidelines.
REFERENCES
- References follow the indicated style
Author Response
We are most grateful to your Reviewer for their diligent comments.
ABSTRACT
- Vitamin E has been added to the Abstract - RESPONSE: thank you!
- Do not use the first person in scientific publications. Applicable to the whole document. RESPONSE: changes have been made throughout the manuscript
INTRODUCTION
- The introduction is brief, it should be justified with more literature and current scientific background on the subject, especially on other studies in which other micronutrients have been studied, such as selenium, vitamin B12, choline and probiotics.
- For example explain in detail the next statement that appears in the results which can be dealt with in depth in this section: "By comparison, at postpartum, maternal meth- 289 ylation gains, which are biologically relevant for cadherin signaling and immune function, were seen at 200 and a loss at 102 CpGs".
RESPONSE: We have reflected on this helpful suggestion. The introduction at 623 words is already quite long. We include 21 references. Our aim was to provide a general overview of the topic for a wide audience. The whole article is now >16,000 words following the suggested amendments. Our feeling is that a detailed explanation of signalling and immune function does not fit so well at the start of a scoping review.
Probiotics do not feature in our manuscript.
MATERIAL AND METHODS
- Divide the methodology into the different sections for a better understanding and approach (eligibility criteria, information sources, risk of bias and synthesis of results). Improve the structure, e.g., include data collection and synthesis and methodological quality in a different section.
- Present key elements of the study design at the beginning of the paper. For example: this literature review was conducted using the informative guidelines for scoping reviews [23]. It was designed following the recommendations of the PRISMA Statement… [CITE].
- It is recommended to describe any efforts to address potential sources of bias. Specify the methods used to assess risk of bias in the included studies.
RESPONSE: We have revised the Methodology as suggested. Adding sections has made this section clearer - thank you.
Reference to PRISMA-ScR as advocated by Tricco et al [23] now appears as the second line in this section.
As mentioned at Line 137, "risk of bias was not assessed. Unlike for systematic reviews, the process of assessing the risk of bias and synthesizing findings from individual studies to generate summary findings is not mandatory [24]."
Selection bias was minimised by using two reviewers working independently. Where there was debate concerning the inclusion of a full text article, this was adjudicated by two nutrition experts.
RESULTS
- Give the total number of included studies and participants and summarise relevant characteristics of studies. There is no table with the main characteristics of the sample size (age, week of gestation, BMI, follow-up period, etc). It is proposed, for example, to create a study description section and include it. RESPONSE: We have reflected on this. Again, in view of the size of the document we would resist adding a further table. Instead, we have added additional text at the start of the Results section to set the scene. We have also added information on age, weeks of gestation, BMI, follow-up (where available) to the current tables.
- Create a study selection section and improve the PRISMA diagram with the reference indicated in the methodology section. RESPONSE: We have added the Tricco et al. reference [23] to the figure.
- Line 181, this section is focused in Vitamina B12 and it is discussed vitamin A? RESONSE: Removed - thank you.
- The information regarding studies [47] and [26] is repeated both in the selenium and HIV section and in the vitamin A and malaria section. The information should only appear once. RESPONSE: At Line 152 in the Results section we write: "Selected publications frequently explore the effects of micronutrient supplementation alone or in combination. Others were found to focus on human pathophysiology (e.g., immunity after vaccination or a disease process such as malaria or HIV). Consequently, a particular study may be mentioned more than once under different headings." The paper by Darling et al. [26] is mentioned in detail in Section 3.1 Vitamin A (Line 173-175). It is also referred to briefly much further on at Line 622-623 in Section 3.10 Micronutrients, malaria and HIV. Our thinking here was to remind the reader going immediately to the later section that more details can be found earlier in the paper. In respect of the paper by Okunade et al. [47] at Line 462-470our think was similar. We accept that the second reference to selenium at Line 658 is too long and have much shortened the text.
- When a table is split into two sheets, the sections must be put back in the header. RESPONSE: Amended
- Various abbreviations are used in the tables which are not explained. They should be clarified in the table footnotes. RESPONSE: we have added these.
- Include in the tables what the study objective was. RESPONSE: THank you, we have amended the tables. It should be noted that the objective were rarely related directly to maternal clinical benefit.
- Add all important harms or unintended effects for each study and micronutrient. RESPONSE: We have amended the tables where possible. Harm to the mother potentially associated with micronutrient supplementation is rarely reported. Again, we are mindful of word count in the tables as raised by Reviewer 1.
.
DISCUSSION
- Provide a brief summary of the limitations of the evidence included in the review (e.g. study risk of bias, inconsistency and imprecision). RESPONSE: We believe that we have done this sufficiently for a scoping review in Section 4,6 at Lines 791-808
- Provide a cautious overall interpretation of the results taking into account the objectives, limitations, multiplicity of analyses, results of similar studies, and other relevant evidence. RESPONSE: We believe that we have done this sufficiently for a scoping review in Section 5 at Lines 809-828. The studies we have included are diverse and our view is that only limited direct comparisons exist relevant to the mother.
- Discuss in detail this statement: "Differences in DNA methylation were also seen 291 in breastfed infants of intervention mothers at 4-6 weeks". Compare DNA methylation with other studies. RESPONSE: The Study by Anderson et al [34] is excellent and covered at Lines 290-307 and again at 747-753. We do not consider, however, that a more detailed discussion about DNA methylation and micronutrients is appropriated in this context.
- Discuss the generalizability (external validity) of each of the sections included in the study. RESPONSE: As mentioned above, the identified papers are diverse in nature and often only address one aspect of the topic. We have added a note at Line 813 questioning 'generalizability'.
- Compare the results obtained with current practice guidelines. RESPONSE: One of our lead authors writes:
This is tricky as practice guidelines vary by country. In the US and Canada Fe is recommended prophylactically in pregnancy, but in UK, Aus and NZ iron is only provided after diagnosis of IDA. Recommendations for folate supplementation also vary by country, especially duration and dose. It also depends on baseline. I supp is recommended in NZ and Aus but not UK.
We have added a comment to our conclusion at Lines 814-816 on international recommendation and variations in practice
Round 2
Reviewer 2 Report
No further comments.